# Structural organization and dynamics of FCHo2 docking on membranes

Fatima El Alaoui[1], Ignacio Casuso[2], David Sanchez-Fuentes[3], Charlotte Arpin-Andre[1], Raissa Rathar[3], Volker Baecker[4], Anna Castro[5], Thierry Lorca[5], Julien Viaud[6], Stéphane Vassilopoulos[7], Adrian Carretero-Genevrier[3], Laura Picas[1]*

[1]Institut de Recherche en Infectiologie de Montpellier (IRIM), CNRS UMR 9004, Université de Montpellier, Montpellier, France; [2]U1067 INSERM, Aix-Marseille Université, Marseille, France; [3]Institut d'Électronique et des Systèmes (IES), CNRS UMR 5214, Université de Montpellier, Montpellier, France; [4]Montpellier Ressources Imagerie, BioCampus Montpellier, CNRS, INSERM, Université de Montpellier, Montpellier, France; [5]Centre de Biologie Cellulaire de Montpellier (CRBM), CNRS UMR UMR 5237, Université de Montpellier, Montpellier, France; [6]INSERM UMR1297, Institute of Metabolic and Cardiovascular Diseases (I2MC), University of Toulouse, Paul Sabatier University, Toulouse, France; [7]Sorbonne Université, INSERM, Institute of Myology, Centre of Research in Myology, UMRS 974, Paris, France

*For correspondence:
laura.picas@irim.cnrs.fr

Competing interest: The authors declare that no competing interests exist.

**Abstract** Clathrin-mediated endocytosis (CME) is a central trafficking pathway in eukaryotic cells regulated by phosphoinositides. The plasma membrane phosphatidylinositol-4,5-bisphosphate (PI(4,5)$P_2$) plays an instrumental role in driving CME initiation. The F-BAR domain-only protein 1 and 2 complex (FCHo1/2) is among the early proteins that reach the plasma membrane, but the exact mechanisms triggering its recruitment remain elusive. Here, we show the molecular dynamics of FCHo2 self-assembly on membranes by combining minimal reconstituted in vitro and cellular systems. Our results indicate that PI(4,5)$P_2$ domains assist FCHo2 docking at specific membrane regions, where it self-assembles into ring-like-shaped protein patches. We show that the binding of FCHo2 on cellular membranes promotes PI(4,5)$P_2$ clustering at the boundary of cargo receptors and that this accumulation enhances clathrin assembly. Thus, our results provide a mechanistic framework that could explain the recruitment of early PI(4,5)$P_2$-interacting proteins at endocytic sites.

## Introduction

The biogenesis of clathrin-coated vesicles requires precise and coordinated recruitment of more than ~50 different proteins to undergo the bending, elongation, and fission of the plasma membrane (*Taylor et al., 2011*; *Haucke and Kozlov, 2018*). Different factors assist in recruiting endocytic proteins, such as the interaction with phosphoinositides (*Posor et al., 2015*), curvature sensing, and protein-protein interactions (*Doherty and McMahon, 2009*). Also, the resistance of the plasma membrane that results from the membrane-cytoskeleton adhesion sets the rate of forming transport vesicles (*Sheetz, 2001*). Although the initiation of endocytosis is a critical step, the exact mechanism triggering the nucleation of endocytic proteins at the plasma membrane is not well understood. The early stages of clathrin-mediated endocytosis (CME) entail the nucleation of adaptor and accessory proteins, cargo, and lipids to undergo the bending of the plasma membrane (*Godlee and Kaksonen, 2013*). Several studies have shown that the Fer/CIP4 homology domain-only protein 1 or 2 (FCHo1/2) is among the early proteins recruited at endocytic sites (*Taylor et al., 2011*; *Henne et al., 2010*), where it establishes a network of interactions with pioneer proteins, such as Eps15, adaptor protein 2 (AP2), and

transmembrane cargo (*Taylor et al., 2011*; *Ma et al., 2016*; *Hollopeter et al., 2014*). FCHo paralogs associate with membranes through the dimerization of F-BAR domains displaying a shallow concave surface that interacts with acidic phospholipids (*Henne et al., 2010*; *Frost et al., 2008*; *Henne et al., 2007*). A polybasic motif follows the F-BAR scaffold and provides a selective recognition for $PI(4,5)P_2$ (*Ma et al., 2016*). Finally, FCHo1/2 is flanked at the C-terminal by a μ-homology domain (μ-HD) that directly binds with multiple early endocytic proteins such as Eps15, intersectin 1, or CALM (*Henne et al., 2010*; *Ma et al., 2016*; *Umasankar et al., 2012*). Indeed, FCHo1/2 is required to recruit Eps15 on membranes (*Day et al., 2021*), and the assembly of a FCHo1/2-Eps15-AP2 complex is essential to drive efficient cargo loading (*Ma et al., 2016*). The recruitment of FCHo1/2 on membranes is central to initiate the endocytic activity, but the underlying molecular mechanism remains unclear.

Here, we combined sub-diffraction microscopy and high-speed atomic force microscopy (HS-AFM) with in vitro and in cellulo reconstituted systems to show that $PI(4,5)P_2$ domains regulate FCHo2 docking on flat membranes, where it self-assembles into ring-like-shaped protein structures that are compatible with the size and temporal scale of CME. Our results indicate that, in the absence of metabolizing enzymes, FCHo2 can engage a local $PI(4,5)P_2$ enrichment at the boundaries of clathrin-regulated cargo receptors and enhance the formation of clathrin-positive assemblies. Finally, manipulation of membrane curvature through lithographic approaches showed that $PI(4,5)P_2$ promotes the partition of FCHo2 at the edges of dome-like structures. Collectively, our work points out $PI(4,5)P_2$ lateral lipid heterogeneities as an organizing mechanism supporting the docking and self-organization of $PI(4,5)P_2$-interacting proteins that, like FCHo2, participate in the initial stages of CME.

## Results
### FCHo1/2 drives $PI(4,5)P_2$ clustering formation on cellular membranes
To study FCHo1/2 recruitment on cellular membranes, we monitored by airyscan microscopy the binding of recombinant full-length FCHo2-Alexa647 on plasma membrane sheets. We generated plasma membrane sheets by ultrasound-mediated unroofing of cells stably expressing the transferrin receptor (TfR-GFP) as a model of cargo receptor regulated by FCHo1/2 (*Henne et al., 2010*). We monitored the subcellular dynamics of lipids using TopFluor fatty acid conjugates, as previously reported (*Zewe et al., 2020*). Thus, we loaded HT1080 unroofed cells with fluorescent $PI(4,5)P_2$ (TF-T-MR-$PI(4,5)P_2$) or phosphatidylserine as a control (PS, TF-TMR-PS) (*Figure 1A and B*). We characterized the steady-state organization of fluorescent $PI(4,5)P_2$ and its ability to form domains as compared to other anionic lipids (*Figure 1—figure supplement 1*), in agreement with previous numerical simulations (*Koldsø et al., 2014*). The functionality of recombinant FCHo2 was determined by performing a tubulation assay using membrane sheets made of brain polar lipids, as previously reported (*Itoh et al., 2005*; *Figure 1—figure supplement 2*). Kymograph analysis on plasma membrane sheets showed that FCHo2 is preferentially recruited to $PI(4,5)P_2$-enriched domains often co-localized with the TfR (*Figure 1—figure supplement 3*). We found that the kinetics of FCHo2 recruitment on these regions was faster than in the absence of $PI(4,5)P_2$ enrichment. To determine if the spatial recruitment of FCHo2 is promoted by $PI(4,5)P_2$, we monitored the protein binding on supported lipid bilayers made of 20% of total negatively charged lipids and different % mol of $PI(4,5)P_2$. As expected, increasing amounts of $PI(4,5)P_2$ favored FCHo2 binding on flat membranes (*Figure 1—figure supplement 4*). The temporal analysis of FCHo2 recruitment also showed that binding to $PI(4,5)P_2$ domains resulted in the formation of long-lived FCHo2 puncta (*Figure 1—figure supplement 5*), whereas its association with a homogenous distribution of $PI(4,5)P_2$ was more likely to lead to FCHo2 disassembly. We observed that the F-BAR domain alone (i.e., without the region rich in positively charged amino acids of the extended F-BAR construct, F-BAR-x, described in *Henne et al., 2007*) also displayed preferential recruitment to $PI(4,5)P_2$-enriched regions, in agreement with the enhanced binding of the domain at 5% mol of $PI(4,5)P_2$ (*Figure 1—figure supplement 4*). Collectively, these results confirm the functionality of full-length FCHo2-Alexa647 and indicate that $PI(4,5)P_2$-enriched regions facilitate FCHo2 docking on membranes.

The formation of $PI(4,5)P_2$ clusters at the plasma membrane orchestrates the recruitment of $PI(4,5)P_2$-binding proteins via ionic-lipid protein interactions (*Honigmann et al., 2013*; *van den Bogaart et al., 2011*). Several structural domains of endocytic proteins, including the F-BAR domain of Syp1, locally accumulate $PI(4,5)P_2$ on in vitro membranes (*Zhao et al., 2013*; *Picas et al., 2014*), and we

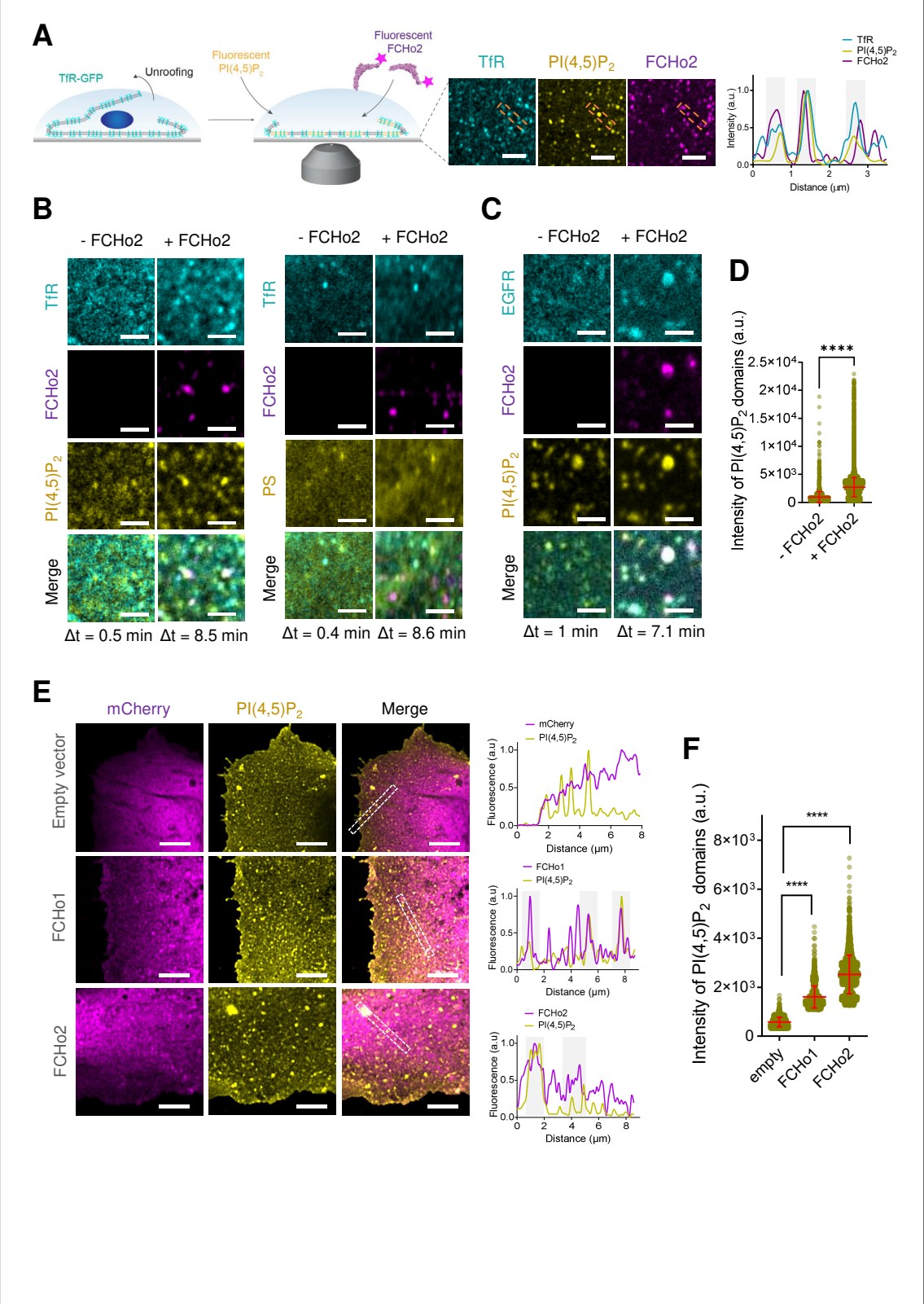

**Figure 1.** FCHo2 induces PI(4,5)P$_2$ clustering at the boundary of cargo receptors. (**A**) Left: cartoon of the experimental setup to monitor the recruitment of FCHo2-Alexa647 on cellular membranes loaded with fluorescent PI(4,5)P$_2$. Right: representative still airyscan images showing the co-localization of TfR-GFP (cyan), PI(4,5)P$_2$ (yellow), and FCHo2 (magenta) on plasma membrane sheets and cross-sectional analysis of FCHo2-positive puncta highlighted by the orange dashed box. Scale bar, 2 μm. (**B, C**) Representative z-projected time-lapse confocal images of plasma membrane sheets showing the

*Figure 1 continued on next page*

*Figure 1 continued*

co-localization of PI(4,5)P$_2$ or PS (yellow) and clathrin-regulated cargoes TfR or EGFR (cyan) before (-FCHo2) and after the addition of FCHo2 (+FCHo2, magenta). The projected time interval (Δt) is specified below each image. Scale bar, 2 μm. (**D**) Distribution of the intensity of PI(4,5)P$_2$ domains before (-FCHo2) and after addition of FCHo2 (+FCHo2) on plasma membrane sheets. Mean ± SD, in red. Welch's *t*-test (****p<0.0001). The number of PI(4,5)P$_2$ domains analyzed from three replicates was n = 4397 and n = 33815, respectively, from at least three replicates. (**E**) Representative airyscan images showing a maximum intensity projection of one stack of the basal plasma membrane of HT1080 cells transiently transfected with either mCherry (empty vector), mCherry-FCHo1, or mCherry-FCHo2 (magenta) and stained for endogenous PI(4,5)P$_2$ (yellow). Cross-sectional analysis of plasma membrane regions highlighted by the white dashed box in the corresponding image. Scale bar, 4 μm. (**F**) Distribution of the intensity of PI(4,5)P$_2$ domains at the plasma membrane of cells transfected with either mCherry (empty), mCherry-FCHo1 (FCHo1), or mCherry-FCHo2 (FCHo2). Mean ± SD, in red. One-way ANOVA (****p<0.0001). The number of PI(4,5)P$_2$ domains analyzed was n = 1647, n = 2,164, and n = 2608, respectively, from three independent experiments.

The online version of this article includes the following source data and figure supplement(s) for figure 1:

**Source data 1.** Intensity of PI(4,5)P$_2$ domains.

**Figure supplement 1.** Representative airyscan images showing the lipid organization of different acidic TF-TMR lipid dyes (PI(4,5)P$_2$, PI(4)P, and phosphatidic acid [PA] in gray) on lipid bilayers containing 20% mol of total negative charge relative to a neutral lipid dye (OG-DHPE or Atto647-DOPE).

**Figure supplement 2.** Recombinant FCHo2 forms membrane tubes in vitro.

**Figure supplement 3.** Recruitment dynamics of FCHo2 on plasma membrane sheets.

**Figure supplement 4.** FCHo2 binding on membranes is assisted by PI(4,5)P$_2$.

**Figure supplement 4—source data 1.** Binding and intensity of FCHo2 and F-BAR.

**Figure supplement 5.** FCHo2 is recruited to PI(4,5)P$_2$-enriched domains.

**Figure supplement 5—source data 1.** Lifetime of FCHo2.

**Figure supplement 6.** Fluorescence quantification over time of the EGFR (cyan), PI(4,5)P$_2$ (yellow) upon injection of 1 μM of FCHo2-Alexa647 (magenta) on EGFR-positive spots on plasma membrane sheets.

showed that BIN1 recruits its downstream partner dynamin through this mechanism. We thus investigated the impact of FCHo2 on PI(4,5)P$_2$ clustering formation. Injection of FCHo2-Alexa647 on TfR-GFP plasma membrane sheets led to the binding of FCHo2 and the formation of sub-micrometric puncta that co-localized with PI(4,5)P$_2$ and the TfR (*Figure 1A*). Analysis of the protein dynamics showed that FCHo2 binding convoyed a redistribution of the PI(4,5)P$_2$ signal on TfR-GFP-positive puncta but not on PS-labeled plasma membrane sheets (*Figure 1B*). To determine if FCHo2-mediated PI(4,5)P$_2$ enrichment was a general feature of FCHo2, we monitored its binding relative to another clathrin-regulated cargo, the EGF receptor (EGFR) (*Figure 1C*). In this case, we observed a concomitant increase of both the PI(4,5)P$_2$ and EGFR signal at FCHo2-positive puncta (*Figure 1C*, *Figure 1— figure supplement 6*), possibly as a result of the interaction of the EGFR juxtamembrane domain with PI(4,5)P$_2$ (*Wang et al., 2014*). Quantification of the intensity of PI(4,5)P$_2$ domains in the absence of other endocytic proteins and ATP to prevent the activation of type I phosphatidylinositol 4-phosphate 5-kinase (*Krauss et al., 2006*) confirmed a local increase in the PI(4,5)P$_2$ signal after the addition of FCHo2 on cellular membranes (*Figure 1D*). Therefore, pointing out that PI(4,5)P$_2$ clustering at the boundary of clathrin-regulated receptors is a direct effect of FCHo2 binding and not due to de novo production of PI(4,5)P$_2$.

To confirm that FCHo1/2 proteins induce PI(4,5)P$_2$ clustering formation in cellulo, we overexpressed FCHo1 and 2 in HT1080 cells. As previously reported, we confirmed that FCHo1/2 localized preferentially at the plasma membrane (*Henne et al., 2010*), organized into protein assemblies of different dimensions (*Figure 1E*). In agreement to what we observed on plasma membrane sheets, our in cellulo data showed a co-localization between FCHo1/2 and endogenous plasma membrane PI(4,5)P$_2$ (see Materials and methods; *Figure 1E*). Furthermore, we detected a significant increase in the average intensity of PI(4,5)P$_2$ domains in the presence of FCHo proteins (*Figure 1F*). Altogether, these results suggest that FCHo1/2 has the capability to accumulate PI(4,5)P$_2$ locally.

## FCHo2-mediated PI(4,5)P$_2$ clustering primes pre-endocytic events

We next analyzed if FCHo2-mediated PI(4,5)P$_2$ clustering participates in the formation of clathrin-coated structures. Both membrane activity and FCHo2 interaction with binding partners are required to direct clathrin-coated structures' growth and stability (*Lehmann et al., 2019*). Whereas the

membrane-bending activity and interaction with negatively charged lipids is primarily encoded within the F-BAR domain (*Henne et al., 2010*), the interaction with AP2 and Eps15 is mediated by a downstream AP2-activating (APA) domain and C-terminal μ-HD (*Ma et al., 2016*; *Hollopeter et al., 2014*). We determined by immunofluorescence the formation of clathrin-positive puncta on supported lipid bilayers made of 20% of negatively charged lipids incubated with nonlabeled F-BAR domain (residues 1–262), extended F-BAR domain (F-BAR-x, residues 1–430) (*Henne et al., 2010*), or full-length FCHo2 (*Figure 2A and B*). We reconstituted clathrin-coat assembly in vitro by using active cytosolic components from *Xenopus* egg extracts supplemented with ATP and GTPγS, as previously reported (*Walrant et al., 2015*; *Daste et al., 2017*). As expected, in the absence of FCHo2, the addition of cytosolic extracts leads to the appearance of clathrin-positive puncta as compared to the ATP and GTPγS alone (*Figure 2C*, *Figure 2—figure supplement 1*). Under these conditions, we could also detect a residual signal of the FCHo2 antibody, corresponding to the endogenous protein present in the cytosolic extracts. Incubation of 1 μM of FCHo2 with cytosolic extracts on 5% PI(4,5) P$_2$-containing lipid bilayers resulted in a twofold increase in the normalized intensity of the PI(4,5)P$_2$ signal on clathrin-positive puncta (*Figure 2C and D*), which is convoyed by a sevenfold increase in the normalized intensity of clathrin-positive spots (*Figure 2C and E*). We observed a moderate increase in the local intensity of both PI(4,5)P$_2$ and clathrin structures in the presence of the F-BAR domain alone, in agreement with its reported ability to assist PI(4,5)P$_2$ microdomain formation (*Zhao et al., 2013*). Furthermore, this effect was intensified by the F-BAR-x, supporting the combined contribution of the APA domain (*Ma et al., 2016*; *Lehmann et al., 2019*) and the enhanced tubulation effect of this construct (*Henne et al., 2010*) in promoting a local PI(4,5)P$_2$ enrichment and the formation of clathrin-positive structures (*Figure 2C–E*). Replacement of PI(4,5)P$_2$ by PS preserved FCHo2 association with membranes but prevented the detection of clathrin in our in vitro assay, supporting the functional role of PI(4,5)P$_2$ in clathrin-coat formation. Thus, these results indicate that, in addition to its membrane-remodeling activity, FCHo2 might promote clathrin assembly by clustering PI(4,5)P$_2$ and that the association with interacting partners in cytosolic extracts enhances this effect, possibly via the APA domain.

## Assembly of FCHo2 into molecular clusters prompts membrane bending

A characteristic hallmark of endocytic proteins on cellular membranes is their spatial organization into punctate structures (*Henne et al., 2010*; *Ma et al., 2016*), and we systematically observed this feature on plasma membrane sheets and in vitro membranes (*Figure 1*, *Figure 1—figure supplement 5*). Co-segregation of early endocytic proteins into sub-micrometer scale clusters relies on multivalent interactions (*Day et al., 2021*). Therefore, we asked what might be the role of PI(4,5)P$_2$ in the spatial organization of FCHo2. We estimated by airyscan microscopy the average size of FCHo2 puncta on cellular membranes (i.e., plasma membrane sheets) and on 5% PI(4,5)P$_2$-containing bilayers after the addition of 1 μM of FCHo2-Alexa647 (*Figure 3A and B*), which was on both cases ~0.067 μm$^2$. On lipid bilayers doped with 5% PI(4,5)P$_2$, we could also detect that the F-BAR domain alone can assemble into punctate structures, although the average size was ~0.094 μm$^2$. Interestingly, replacing 5% PI(4,5)P$_2$ by PS led to a homogeneous surface distribution of FCHo2 and prevented the detection of sub-micrometric puncta. Therefore, suggesting that PI(4,5)P$_2$ multivalent interactions through the F-BAR promote FCHo2 segregation into molecular clusters.

Next, we used atomic force microscopy (AFM) to investigate FCHo2 molecular clusters at nanometer-scale resolution. Supported lipid bilayers doped with 5% PI(4,5)P$_2$ were formed on freshly cleaved mica disks (see Materials and methods). Before adding proteins, we confirmed the homogeneity and absence of defects of supported bilayers under the imaging buffer. Injection of full-length FCHo2 at 1 μM in the imaging chamber resulted in sub-micrometric protein patches with a median dimension of ~0.005 μm$^2$ that protruded out of the flat membrane surface with an average height of ~47 ± 3 nm (*Figure 3C–E*). An increase in the setpoint force from minimal values (a few tens of pN) to intense forces (around 100 pN) resulted in the reduction of the height of FCHo2 clusters down to ~15 ± 2 nm, thus suggesting that FCHo2 can moderately bend supported lipid membranes at minimal AFM imaging forces.

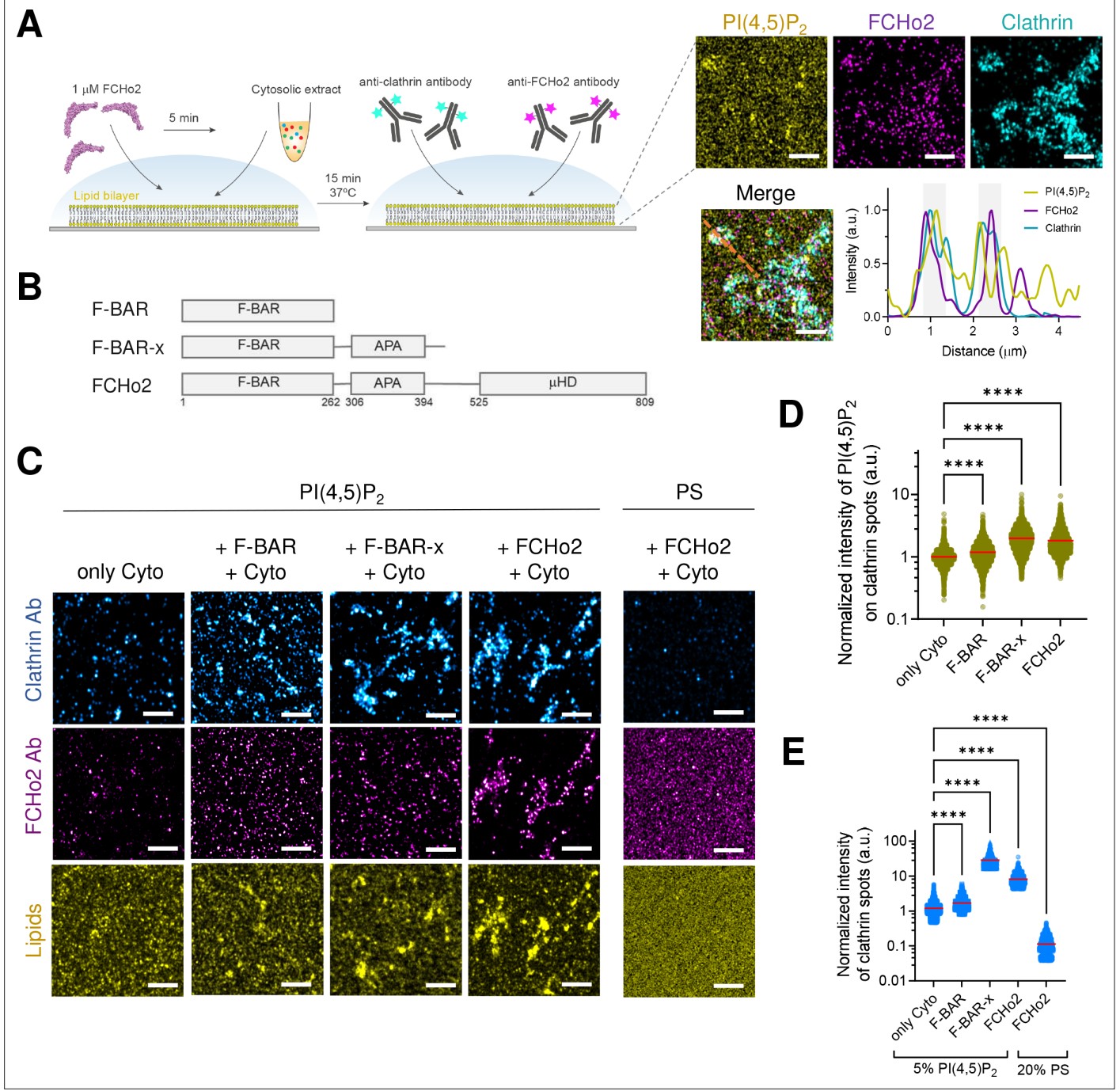

**Figure 2.** FCHo2 induces PI(4,5)P₂ clustering on clathrin-positive puncta. (**A**) Left: cartoon of the in vitro clathrin-coat assembly with cytosolic extracts and immunofluorescence assay to detect FCHo2 and clathrin-positive structures on 5% mol PI(4,5)$P_2$-containing supported lipid bilayers. Left: representative still airyscan images on lipid bilayers doped with fluorescent PI(4,5)$P_2$ (yellow) showing the co-localization with FCHo2 (magenta) and clathrin (cyan). Intensity profile along the orange dotted line in the corresponding image. Scale bar, 2 μm. (**B**) Protein domain mapping of F-BAR domain (residues 1–262), F-BAR-x (residues 1–430), or full-length FCHo2 showing the F-BAR, APA, and C-terminal μ-HD domains. (**C**) Representative airyscan images of lipid bilayers containing either 5% mol PI(4,5)$P_2$ or 20% PS (yellow) (total negative charge is 20%) that were incubated with recombinant FCHo2 (+FCHo2), F-BAR domain (+F-BAR), or F-BAR-x domain (+F-BAR-x) and 4 mg/ml of cytosolic extracts (+Cyto) or only with the cytosolic extract (only Cyto). Clathrin (blue) and FCHo2 (magenta) were detected by immunofluorescence (antibody, Ab). Scale bar, 2 μm. (**D, E**) Log10 scale distribution of the normalized intensity of fluorescent PI(4,5)$P_2$ on clathrin-positive structures and normalized intensity of clathrin structures, respectively, in the absence (only Cyto) or presence of either FCHo2, F-BAR, or F-BAR-x combined to cytosolic extracts on PI(4,5)$P_2$ or PS-containing bilayers. Mean, in red. One-way ANOVA (****p<0.0001). The number of clathrin-positive spots analyzed was n = 11,564, n = 9292, n = 4719, n = 2205, and n = 1635, as set in the graph from at least three independent experiments.

*Figure 2 continued on next page*

*Figure 2 continued*

The online version of this article includes the following source data and figure supplement(s) for figure 2:

**Source data 1.** Normalized intensity of PI(4,5)P$_2$ and intensity of clathrin spots.

**Figure supplement 1.** Representative airyscan images of the immunofluorescence assay showing the localization of PI(4,5)P$_2$ (gray), FCHo2 (magenta), and clathrin (cyan, anti-clathrin antibody) on 5% PI(4,5)P$_2$-containing lipid bilayers incubated with ATP and GTPγS (+energy) or with the cytosolic extract and energy mix (+cytosol).

**Figure supplement 1—source data 1.** Intensity of clathrin spots.

## FCHo2 forms ring-like-shaped assemblies on flat membranes

To establish the biogenesis of FCHo2 clusters at the molecular level, we used HS-AFM. Real-time imaging of the initial stages revealed the entire molecular process of FCHo2 cluster formation, from the binding of single FCHo2 homodimers to the growth of molecular clusters (*Figure 4A*). Representative time-lapse images and kymograph analysis along the dashed region at t = 0 s showed that the

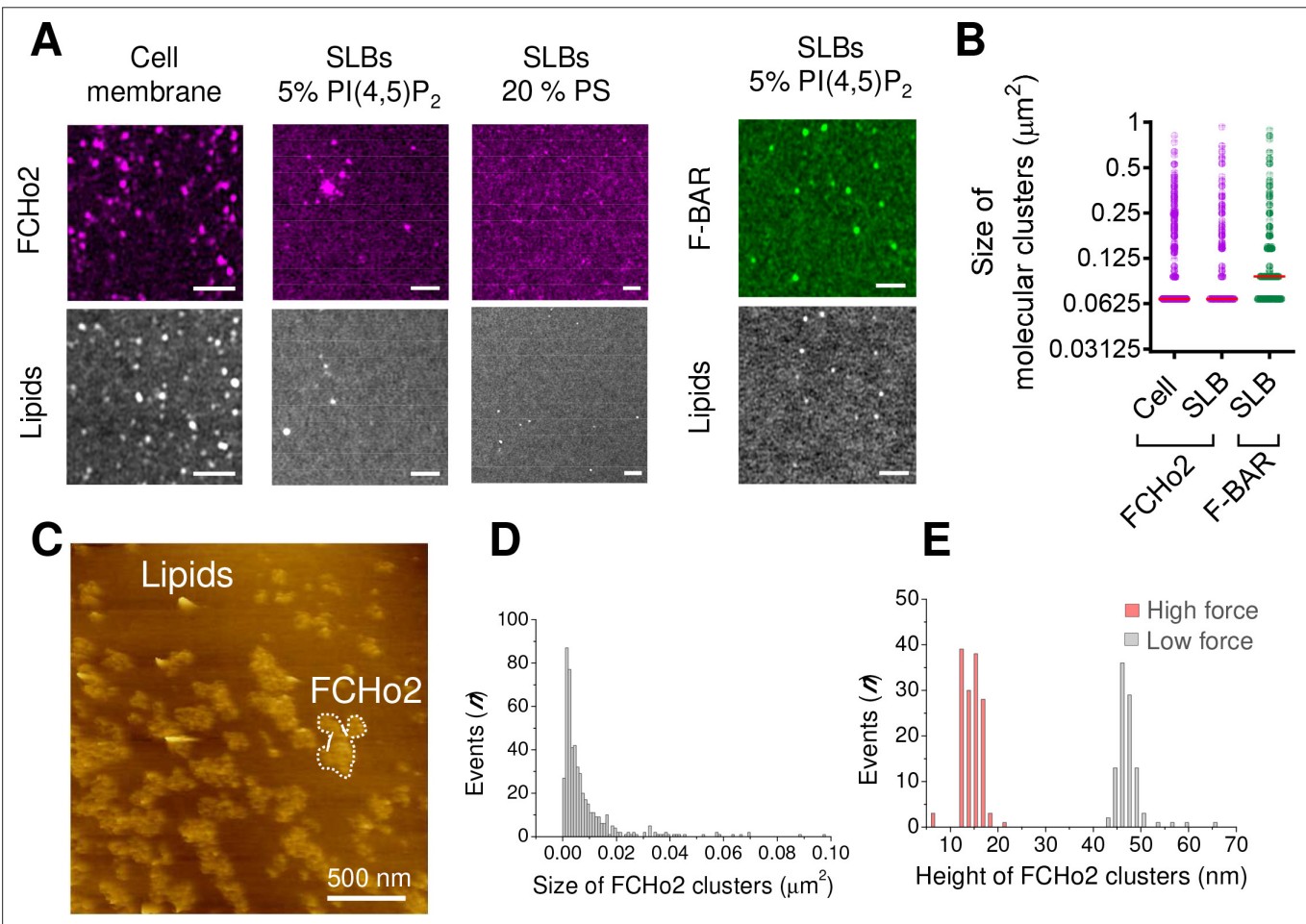

**Figure 3.** FCHo2 forms molecular clusters on PI(4,5)P$_2$-containing bilayers. (**A**) Representative airyscan images of plasma membrane sheets or lipid bilayers doped with either 5% mol PI(4,5)P$_2$ or 20% PS (total negative charge is 20%) and incubated with FCHo2-Alexa647 (magenta) or F-BAR-Alexa647 (green). Scale bar, 2 µm. (**B**) Size distribution of FCHo2 (magenta) and F-BAR (green) molecular clusters (in µm$^2$) on plasma membrane sheets (cell) or lipid bilayers (SLB) containing 5% mol of PI(4,5)P$_2$. Median value is displayed in red. The number of molecular clusters analyzed was n = 10,793, n = 19,287, and n = 7028, as set in the graph from at least three independent experiments. (**C**) Representative atomic force microscopy (AFM) image of FCHo2 molecular clusters (white dashed region) on supported lipid bilayers containing 5% mol PI(4,5)P$_2$. (**D**) Size distribution of FCHo2 molecular clusters (in µm$^2$) obtained from AFM images. (**E**) Height distribution of FCHo2 molecular clusters (in nm) at low (gray) and high (red) setpoint forces from at least three replicates.

The online version of this article includes the following source data for figure 3:

**Source data 1.** Size and height of molecular clusters.

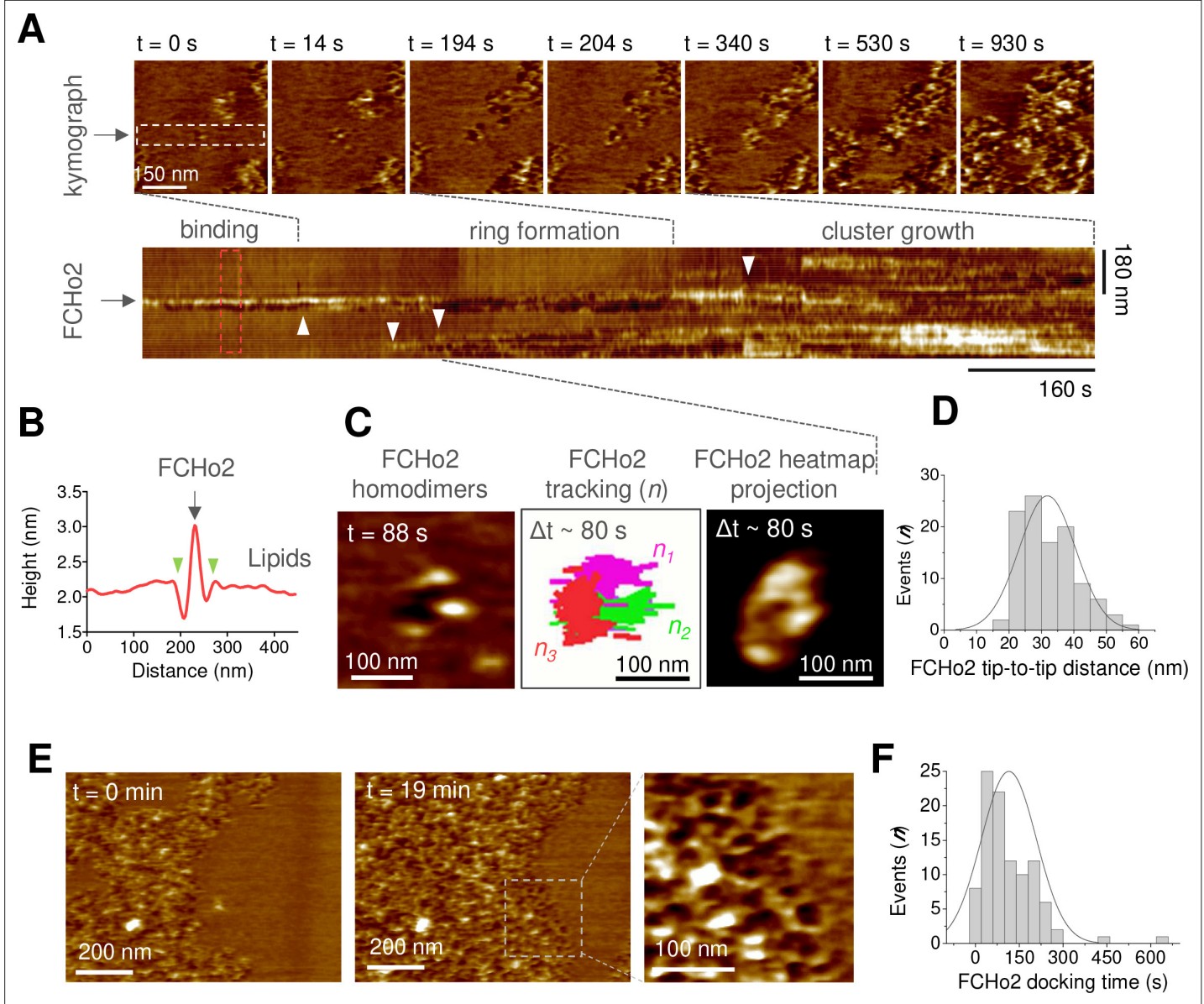

**Figure 4.** Molecular dynamics of FCHo2 self-assembly on PI(4,5)P$_2$-containing membranes. (**A**) High-speed atomic force microscopy (HS-AFM) movie frames of the binding of FCHo2 on 5% mol PI(4,5)P$_2$-containing membranes. Kymograph analysis performed on the white outline to display the representative stages of FCHo2 binding and self-assembly on flat membranes. White arrowheads highlight the docking of new FCHo2 homodimers to the growing molecular cluster. (**B**) Profile analysis along the red outline in (**A**). Green arrowheads highlight the membrane invagination upon binding of FCHo2 on lipid membranes. (**C**) HS-AFM movie snapshot of the FCHo2 ring formation along with a representative tracking of individual homodimers (n = 3) and the heat map projection within a time interval, Δt, of ~80 s. (**D**) Size distribution (in nm) estimated from individual FCHo2 proteins before the cluster growth (t ≤ 200 s). (**E**) HS-AFM snapshots (t = 0 min and t = 19 min) illustrating the growth of FCHo2 molecular clusters into ring-like-shaped protein patches. Magnified image corresponds to the dashed outline. (**F**) Distribution of the docking time (in s) of individual and ring-like FCHo2 assemblies during the cluster growth from at least three replicates.

The online version of this article includes the following source data and figure supplement(s) for figure 4:

**Figure supplement 1.** Representative high-speed atomic force microscopy (HS-AFM) frames of FCHo2 assembly on 5% PI(4,5)P$_2$-containing lipid bilayers.

**Source data 1.** FCHo2 distance and docking time.

binding of individual FCHo2 proteins engages an indentation of few nanometers in the lipid membrane adjacent to the protein surface (**Figure 4B**, green arrowheads), as delineated from the cross-sectional profile along the red dashed box in the corresponding kymograph (**Figure 4A**). The binding of FCHo2 was rapidly prompted by the arrival of additional FCHo2 homodimers (as highlighted by white arrowheads in the kymograph). This stage of the process was characterized by minimal lateral interactions and the existence of contacts between adjacent FCHo2 homodimers, ultimately leading to a ring-like organization (**Figure 4C**, **Figure 4—figure supplement 1**). This dynamic reorganization, which we named as the 'ring formation' step, spanned over ~80–100 s.

The identification of individual proteins at the initial stages allowed us to extract the average dimension of the full-length protein interacting with the flat membrane, which was ~32 ± 8 nm (**Figure 4D**) and in good agreement with the size of the F-BAR domain reported from electron microscopy micrographs (**Frost et al., 2008**). After the ring formation, we observed the growth of FCHo2 clusters through docking events that involved individual FCHo2 homodimers and the coalescence of adjacent FCHo2 rings (**Figure 4A**, white arrowheads). FCHo2 self-organization into hollow ring-like assemblies was particularly discernible at the growth front of FCHo2 molecular clusters (**Figure 4E** and magnified image). Although the entire formation of FCHo2 molecular clusters expanded over few tens of minutes, we found that the docking of individual proteins and rings to support the expansion of the cluster took place every ~115 ± 94 s (**Figure 4F**). Collectively, our results suggest that on flat membranes FCHo2 exhibits an intrinsic ability to self-assemble into a ring-like-shaped molecular complex independently of the local protein density.

## PI(4,5)P$_2$ assists FCHo2 partitioning on curved membranes

Because the transition from a flat surface to a dome-like invagination is a major step in the formation of clathrin-coated structures (**Haucke and Kozlov, 2018**), we set out to monitor the organization of FCHo2 on curved membranes. To this end, we engineered arrays of SiO$_2$ vertical nano-domes of radii $R$ ~ 150 nm using soft nano-imprint lithography (soft-NIL) (**Figure 5A**), as previously reported (**Sansen et al., 2020**). We functionalized SiO$_2$ nano-patterned substrates with supported lipid bilayers containing 20% of negatively charged lipids. Curvature sensing abilities of F-BAR proteins rely on hydrophobic insertion motifs, as in the case of syndapin 1 (**Ramesh et al., 2013**), or intrinsically disordered regions (IDRs) as reported for FBP17 (**Su et al., 2020**). Indeed, the F-BAR domain of FBP17 displays minimal curvature sensing properties in vitro as compared to its IDR. Thus, we determined by airyscan microscopy the surface organization of the F-BAR domain and full-length FCHo2 labeled with Alexa647 on nano-domes in the presence of lipid bilayers containing 5% mol of PI(4,5)P$_2$ (**Figure 5B**). The three-dimensional (3D) rendering of the protein signal relative to a reference marker to depict the nano-dome topography (DHPE lipid, in gray) showed that while the F-BAR domain is excluded from the nanostructure, the FCHo2 staining is well present at the base of the nano-dome (**Figure 5B**). We hypothesized that if the association of FCHo2 with PI(4,5)P$_2$ is essential for its localization on curved membranes, disrupting this interaction would change its spatial organization. Indeed, this was the case, and replacing PI(4,5)P$_2$ by PS (20% mol PS) led to the complete distribution of FCHo2 all over the nano-dome surface (**Figure 5B**, yellow).

To determine the precise localization of FCHo2 on nano-domes, we performed airyscan acquisitions at two z planes, at the top of the nano-dome and the bottom, according to the z-axis resolution of the setup (~0.35 µm) (**Huff et al., 2017**; **Figure 5C**). First, we determined whether the distribution of PI(4,5)P$_2$ on the nano-domes was homogeneous. To this end, we monitored the axial localization of the PI(4,5)P$_2$ signal relative to the DHPE lipid dye on nano-domes coated with 5% PI(4,5)P$_2$-containing lipid bilayer (**Figure 5C**). The cross-sectional analysis showed an equivalent surface distribution of the PI(4,5)P$_2$ and DHPE lipid dye under our experimental conditions. Second, to confirm that TF-TMR-PI(4,5)P$_2$ signal was replicating the actual organization of the total pool of PI(4,5)P$_2$ on nano-domes, we analyzed the distribution of the PH domain of PLCδ1 PH(PLCδ1), which is a well-established reporter of PI(4,5)P$_2$ (**Lemmon et al., 1995**). As expected, we obtained a homogenous organization of the PH(PLCδ1), as indicated by the detection of the domain signal both on the flat surface (bottom plane) and all over the nano-dome structure (top plane) (**Figure 5D**). Following the same rationale, we monitored the surface localization of the F-BAR and extended F-BAR-x domains, and FCHo2 relative to DHPE, as a reference of the nano-dome height. On nano-domes functionalized with PI(4,5)P$_2$-containing membranes, the F-BAR and F-BAR-x preferentially bind to the flat surface and appear

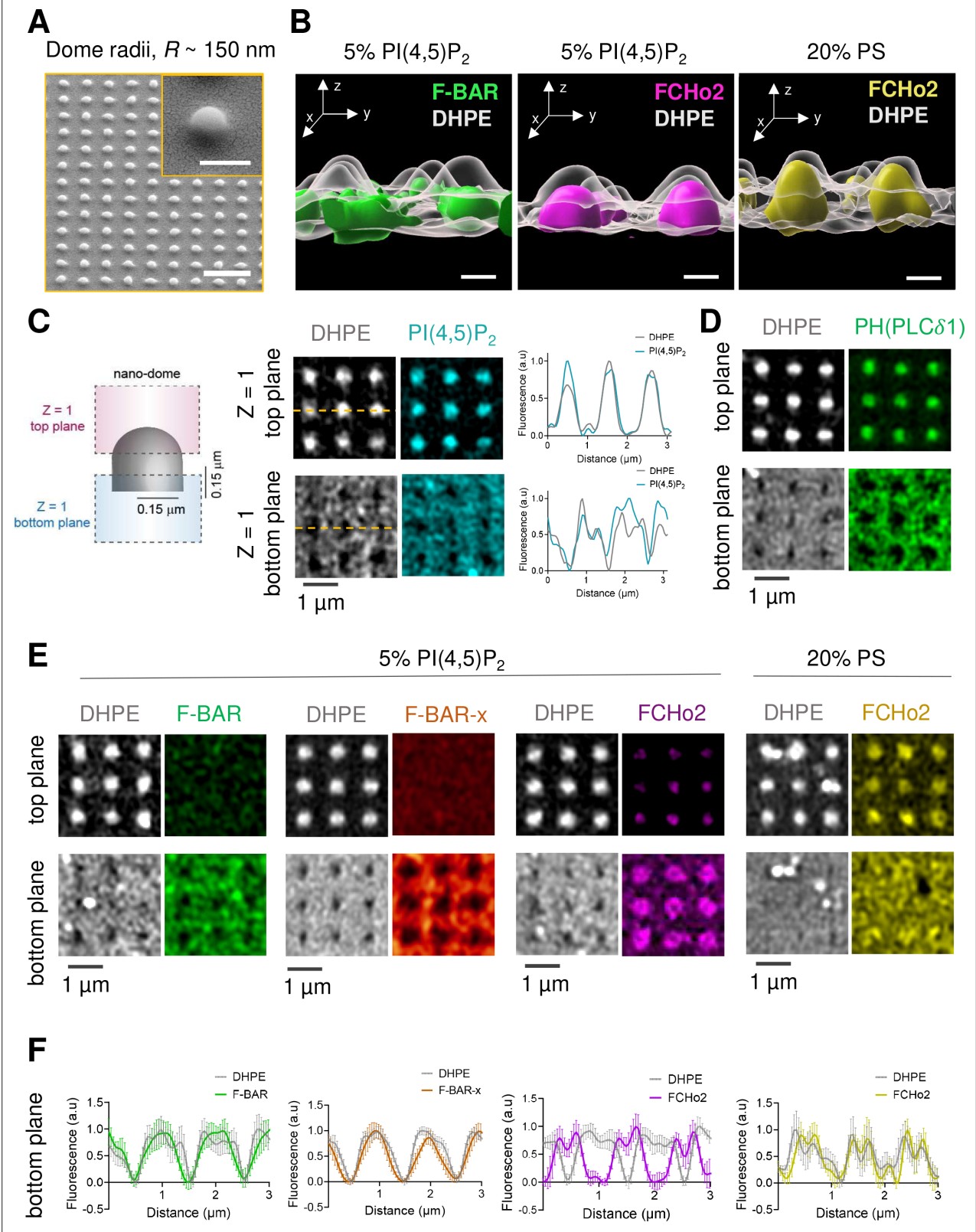

**Figure 5.** PI(4,5)P$_2$ assists the organization of FCHo2 on curved membranes. (**A**) SEM images of SiO$_2$ substrates displaying an array of nano-domes with a radius, $R \sim 150$ nm generated by soft-NIL. Scale bar, 2 µm. Inset, scale bar is 500 nm. (**B**) Representative 3D renders of the surface organization of the F-BAR domain (green) and FCHo2 (magenta) on nano-domes functionalized with either 5% mol PI(4,5)P$_2$ or 20% PS-containing membranes (yellow) relative to the DHPE lipid dye signal (gray). Scale bar, 400 nm. (**C**) Left: schematic representation of the acquisition of a single plane (Z = 1) at the top

*Figure 5 continued on next page*

*Figure 5 continued*

and bottom of nano-domes to visualize the preferential distribution of proteins. Right: representative airyscan images showing the surface distribution of fluorescent DHPE (gray) and PI(4,5)P$_2$ (cyan) on nano-domes along with a cross section (orange dashed line) of the normalized fluorescence intensity of DHPE (gray) and PI(4,5)P$_2$ (cyan) at the top and bottom of the nano-dome. (**D**) Representative airyscan images showing the surface distribution of the PH (PLCδ1) domain (green) on nano-domes functionalized with 5% mol of PI(4,5)P$_2$. (**E, F**) Representative airyscan images (**E**) and average profile analysis of the normalized fluorescence intensity (**F**) showing the surface distribution of fluorescent DHPE (gray) and F-BAR (green), F-BAR-x (orange), and FCHo2 (magenta) on PI(4,5)P$_2$-containing membranes and FCHo2 (yellow) on PS-containing membranes at the bottom of nano-domes. Each curve represents the mean ± SD of n = 20 nano-domes from at least three replicates.

absent from the nano-dome structure (***Figure 5E and F***, green and orange). Whereas FCHo2 revealed a preferential accumulation at the base of the nano-dome (***Figure 5E and F***, magenta). Finally, in the presence of PS membranes, FCHo2 was no longer accumulated at the edges of nano-domes and displayed a homogenous distribution (***Figure 5E and F***, yellow). Collectively, these data suggest that the F-BAR domain of FCHo2 displays minimal sensing of curvatures of radii ~150 nm in vitro and that PI(4,5)P$_2$-mediated accumulation of FCHo2 at the rims of nano-domes is likely to originate via its downstream protein regions (i.e., the disordered central region [***Day et al., 2021***] and C-terminal µ-HD [***Ma et al., 2016***; ***Hollopeter et al., 2014***]), in agreement with the curvature-sensing profile reported for the F-BAR domain protein FBP17 (***Su et al., 2020***).

## Discussion

This study reports a molecular visualization of the docking and self-assembly of the endocytic protein FCHo2 on in vitro and cellular membranes (***Figure 6***). Our results show that PI(4,5)P$_2$ is a primary spatial regulator of the recruitment of FCHo2 by promoting its accumulation and sorting on flat and

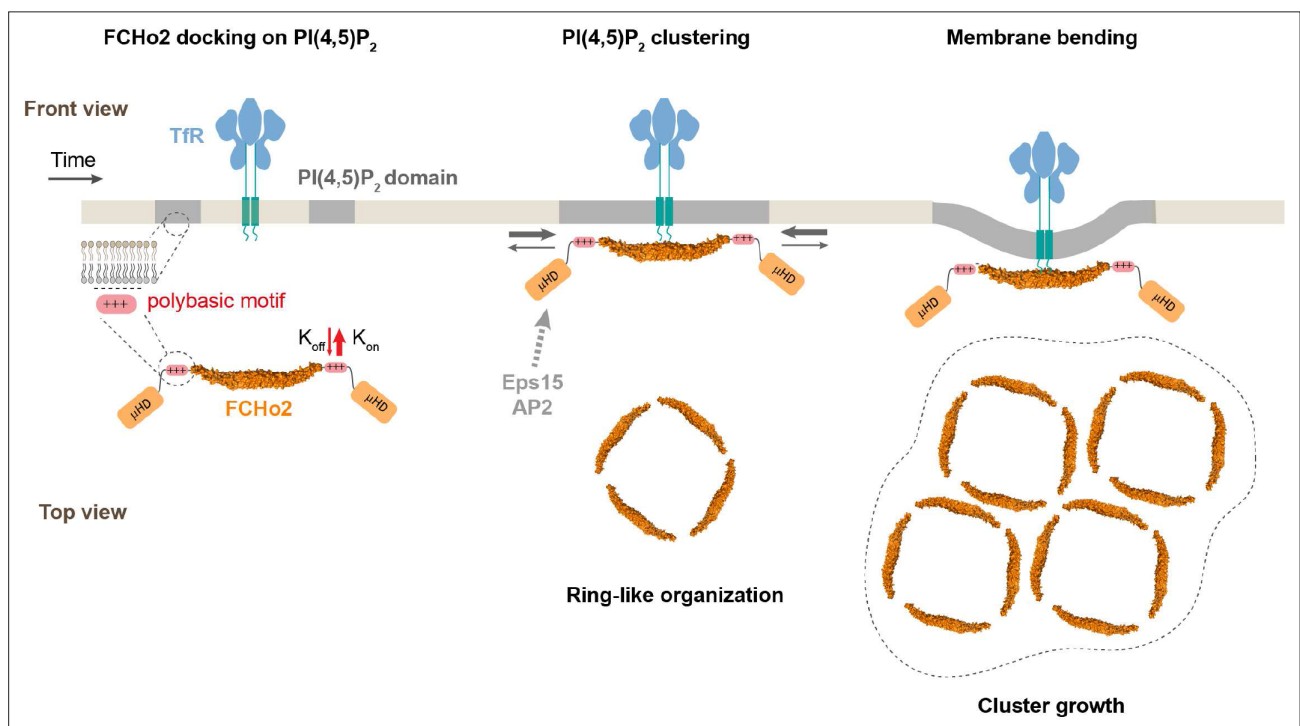

**Figure 6.** Model of FCHo2 docking and self-assembly on membranes. FCHo2 recruitment of membranes is mediated by PI(4,5)P$_2$. Increase in the local PI(4,5)P$_2$ concentration enhances the spatial accumulation of FCHo2 through multivalent interactions between the positively charged surface of the F-BAR domain and polybasic region with PI(4,5)P$_2$ molecules. Binding of FCHo2 on cellular membranes is convoyed by PI(4,5)P$_2$ clustering formation at the boundary of clathrin-regulated cargo receptors and the self-assembly of FCHo2 into a ring-like-shaped protein complex. As a result of this singular organization, the local accumulation of PI(4,5)P$_2$ and APA/µHD of FCHo2 is likely to facilitate the formation of an interacting network with downstream partners, such as Eps15 and AP2 (***Ma et al., 2016***; ***Henne et al., 2007***), ultimately leading to the assembly of clathrin structures on a PI(4,5)P$_2$-rich interface that could be amplified by phosphoinositides metabolizing enzymes (***Posor et al., 2015***). In the absence of endocytic partners, FCHo2 rings grow into sub-micrometric molecular clusters, leading to membrane bending and the partitioning on curved membranes.

curved membranes (*Figures 1 and 6*). These observations support the model that, in addition to protein-protein interactions (*Day et al., 2021*), multivalent lipid-protein interactions play an instrumental role in upholding the early stages of endocytosis. Furthermore, we show that, in the absence of ATP, FCHo2 oligomerization induces PI(4,5)P$_2$ clustering formation on cellular membranes that are often co-localized with TfR and EGFR-positive puncta (*Figure 1*). This association was particularly remarkable in the case of the EGFR and agreed with the observation that electrostatic interaction of the polybasic motifs at the cytoplasmic tail of the EGFR mediates its clustering on PI(4,5)P$_2$-enriched domains (*Koldsø et al., 2014*; *Wang et al., 2014*; *Abd Halim et al., 2015*). As previously reported, FCHo1/2 is needed to recruit Eps15 (*Henne et al., 2010*) and form FCHo1/2-Eps15 micrometer-scale domains on membranes (*Day et al., 2021*). By driving PI(4,5)P$_2$ clustering formation at the boundary of cargo receptors, FCHo2 is likely to improve the stability of a network of PI(4,5)P$_2$-interacting proteins through avidity (*Chen et al., 2019*). Indeed, the F-BAR domain alone was reported to facilitate local PI(4,5)P$_2$ accumulation and our observations confirm a 1.4-fold increase in the formation of clathrin-positive assemblies on in vitro membranes (*Figure 2*). This effect is intensified by the coupled action of membrane tubulation and the APA domain, possibly via AP2 activation, in agreement with previous studies pointing that clustering of Fcho1/2Eps15/AP2 primes endocytosis (*Ma et al., 2016*; *Lehmann et al., 2019*). Furthermore, we discerned clathrin-positive puncta in lower FCHo2 concentrations but not in PI(4,5)P$_2$-depleted membranes, which agrees with the observation that AP2 can create its local pool of PI(4,5)P$_2$ (*Krauss et al., 2006*), although it requires PI(4,5)P$_2$ for its localization and activation at the plasma membrane (*Kadlecova et al., 2017*; *Höning et al., 2005*). Thus, our measurements point out that local PI(4,5)P$_2$ enrichment induced by FCHo2 might operate as a complementary and/or synergistic mechanism to PI(4,5)P$_2$ synthesis on promoting pre-endocytic events (*Figures 1 and 2*).

This work provides the first evidence that FCHo2 self-assembles into ring-like molecular complexes on flat lipid bilayers (*Figures 3 and 4*) and supports the observation of FCHo2 rings at the edges of nascent clathrin-coated structures in living cells (*Lehmann et al., 2019*). This singular organization agrees with a side-lying conformation proposed for F-BAR scaffolds at low protein densities on flat surfaces (*Frost et al., 2008*) and the partitioning of FCHo1/2 at the rims of flat clathrin lattices (*Sochacki et al., 2017*). HS-AFM movies show that the ring formation process is relatively fast and takes place within less than 100 s (*Figure 4*), which is compatible with the temporal scales reported during clathrin-coat assembly (*Taylor et al., 2011*). Previous works reported that lateral contacts stabilize the self-assembly of F-BAR domains on membrane tubules (*Frost et al., 2008*; *Mim et al., 2012*), and our investigations point out that this type of interaction might also occur on flat surfaces. We observed the anisotropic growth and formation of FCHo2 molecular clusters in the absence of other endocytic proteins and, importantly, show that F-BAR proteins can moderately bend flat membranes at high protein densities.

## Conclusions

In conclusion, our study provides a molecular picture of the recruitment and self-assembly of the early endocytic protein FCHo1/2 on membranes. We found that the binding of FCHo2 on cellular membranes promotes the local accumulation of PI(4,5)P$_2$ at the vicinity of clathrin-regulated cargo receptors. As a result, in the absence of phosphoinositides-metabolizing enzymes, FCHo2 can enhance the formation of clathrin structures through PI(4,5)P$_2$-rich interfaces, which could explain previous studies showing that FCHo1/2 depletion slows down the progression of cargo-loaded clathrin structures (*Ma et al., 2016*; *Hollopeter et al., 2014*; *Umasankar et al., 2012*; *Mulkearns and Cooper, 2012*). Because FCHo1/2 is among the first proteins recruited at endocytic sites, the discovery that FCHo2 self-assembles into rings convoyed by local PI(4,5)P$_2$ accumulation and membrane bending provides a fundamental understanding of the initiating mechanism of CME (*Haucke and Kozlov, 2018*; *Lehmann et al., 2019*).

## Materials and methods
### Lipids and reagents
Natural and synthetic phospholipids, including POPC, POPS, Egg-PC, Brain-PS, Brain-PI(4,5)P$_2$, and fluorescent TopFluor-TMR-PI(4,5)P$_2$ and TopFluor-TMR-PS, are from Avanti Polar Lipids, Inc Oregon green 488-DHPE and Alexa Fluor 647 Maleimide labeling kit are from Invitrogen. Atto647N-DOPE

was from Sigma. Monoclonal mouse anti-clathrin heavy chain (dilution 1:1000; Cat# 610499) was from BD Biosciences, and polyclonal rabbit anti-FCHo2 (dilution 1:1000; Cat# NBP2-32694) was from Novusbio.

## Mammalian cell line

HT1080 cells were kindly provided by Dr. N. Arhel, IRIM, CNRS UMR9004, Montpellier, France. Cell lines were verified to be free of mycoplasma contamination, and the identities were authenticated by short tandem repeat (STR) profiling (Eurofins Genomics).

## EGFR-GFP and TfR-GFP stable cell lines

pRRL.sin.cPPT.SFFV-EGFP/IRES-puro was kindly provided by C. Goujon (IRIM, CNRS UMR9004, Montpellier, France), the EGFR-GFP vector was a gift from Alexander Sorkin (Addgene plasmid #32751), and the pBa.TfR.GFP vector was a gift from Gary Banker and Marvin Bentley (Addgene plasmid # 4506). The GFP was replaced by a GFP-delta-ATG using these primers: 5′-gtatatatatGGATCCGT GAGCAAGGGCGAGGAG-3′ and 5′-CTCACATTGCCAAAAGACG-3′. GFP-delta-ATG fragment replaced the GFP- fragment in pRRL.sin.cPPT.SFFV-EGFP/IRES-puro using a BamHI-XhoI digestion. EGFR was amplified with these primers: 5′-caaatatttgcggccgcATGCGACCCTCCGGGACG-3′ and 5′-gtataccggttgaacctccgccTGCTCCAATAAATTCACTGCTTTGTGG-3′ and cloned into pRRL.sin.cPPT. SFFV-EGFP-delta-ATG/IRES-puro using NotI-AgeI to generate a fused EGFR-GFP protein.

Lentiviral vector stocks were obtained by polyethylenimine (PEI)-mediated multiple transfection of 293T cells in six-well plates with vectors expressing Gag-Pol (8.91), the mini-viral genome (pRRL. sin.cPPT.SFFV-EGFR-GFP/IRES-puro), and the Env glycoprotein of VSV (pMD.G) at a ratio of 1:1:0.5. The culture medium was changed 6 hr post transfection and lentivectors containing supernatants harvested 48 hr later, filtered, and stored at –80°C.

The mini-viral genome (pRRL.sin.cPPT.SFFV-TFR-GFP/IRES-puro) and the corresponding lentivectors were generated as detailed in the case of the EGFR-GFP.

To generate a stable cell line HT1080 expressing EGFR-GFP or TfR-GFP, HT1080 cells were transduced in six-well plates using the supernatant of one six-well plates of the lentiviral stock production detailed above. The culture medium was changed 6 hr post transduction. Puromycin (1 µg/ml) was added 48 hr after transduction. The percentage of GFP-expressing cells was enumerated by flow cytometry 72 hr after selection under puromycin.

HT1080 cells constitutively expressing the EGFR-GFP or TfR-GFP were cultured in DMEM GlutaMAX supplemented with 10% fetal calf serum, 100 U/ml of penicillin and streptomycin and 1 µg/ml of puromycin at 37°C in 5% $CO_2$. Cell lines were tested negative for mycoplasma.

## Protein purification and protein labeling

pGEX-6P-1 vectors coding for the mouse full-length FCHo2 (aa 1–809), F-BAR domain (aa 1–262), and F-BAR-x (aa 1–430) were obtained from HT McMahon (MRC Laboratory of Molecular Biology, Cambridge, UK). Proteins were subcloned into a pET28a vector with a PreScission protease cleaving site. Proteins were expressed in BL21(DE3) bacteria and purified by affinity chromatography using a HiTrap chelating column (GE Healthcare) according to the manufacturer's instructions in 50 mM Tris at pH 8.0, 100 mM NaCl. Proteins were expressed overnight at 18°C using 1 mM IPTG. Proteins were then dialyzed overnight in a Slide-A-Lyzer dialysis cassette (MWCO 10000) before Alexa Fluor 647 maleimide labeling following the protocol described by the manufacturer (Invitrogen). Protein concentrations were measured using a Bradford assay (Bio-Rad).

Recombinant GST-eGFP-PH-domain (PLCδ1) detecting $PI(4,5)P_2$ was purified as described in *Sansen et al., 2020*.

## *Xenopus laevis* egg extracts

Laid eggs were rinsed twice in XB Buffer (100 mM KCl, 1 mM $MgCl_2$, 0.1 mM $CaCl_2$, 50 mM sucrose, and 10 mM HEPES at pH 7.7) and subsequently dejellied with 2% cysteine solution pH 7.8. Once dejellied, they were extensively rinsed with XB buffer to completely eliminate cysteine solution.

Eggs were then recovered from a Petri dish and treated with $Ca^{2+}$ ionophore (final concentration 2 µg/ml), and 35 min later, they were crushed by centrifugation for 20 min at 10,000 × *g* at 4°C. The cytoplasmic layer was collected and supplemented with cytochalasin B (50 µg/ml), aprotinin (5 µg/ml),

leupeptin (5 µg/ml), and 10 mM creatine phosphate. Cytoplasmic extract was centrifuged again for 20 min at 10,000 × *g*. Extracts were frozen and then used as described in *Figure 2*.

The energy mix consisted of 1.5 mM ATP, 0.15 mM GTPγS, 16.7 mM creatine phosphate, and creatine phosphokinase 16.7 U/ml, as previously reported (*Wu et al., 2010*).

## Supported lipid bilayers

Lipid mixtures consisted of 80–85% Egg-PC, 10–15% Brain-PS, and 5–10% of Brain-PI(4,5)$P_2$. The amount of total negatively charged lipids was kept to 20% for any of the mixtures containing phosphoinositides at the expenses of Brain-PS. If needed, fluorescent lipids were added to 0.2%.

For fluorescence microscopy experiments, supported lipid bilayers were prepared as described in *Braunger et al., 2013*. Experiments were performed by injecting 20 µl of buffer (20 mM Tris, pH 7.4, 150 mM NaCl, and 0.5 mg/ml of casein). Supported lipid bilayers were imaged on a Zeiss LSM880 confocal microscope.

For HS-AFM experiments, supported lipid bilayers were prepared following the method described in *Picas et al., 2010*. Briefly, large unilamellar vesicles (LUVs, diameter ~100 nm) were obtained by extrusion of multilamellar vesicles of 85% POPC, 10% POPS, and 5% Brain-PI(4,5)$P_2$ in 20 mM HEPES, pH 7.4, 150 mM NaCl. LUVs were supplemented with 20 mM of $CaCl_2$ and deposited onto freshly cleaved mica disks. Samples were incubated for 20 min at 60°C and extensively rinsed with 20 mM HEPES, pH 7.4, 150 mM NaCl, 20 mM EDTA. Finally, bilayers were rinsed and keep under the imaging buffer, 20 mM HEPES, pH 7.4, 150 mM NaCl.

## Plasma membrane sheets

Unroofing of HT1080 cells stably expressing TfR-GFP or EGFR-GFP was performed by tip sonication as reported in *Heuser, 2000*. Cells were rinsed three times in cold Ringer buffer supplemented with $Ca^{2+}$ (155 mM NaCl, 3 mM KCl, 3 mM $NaH_2PO_4$, 5 mM HEPES, 10 mM glucose, 2 mM $CaCl_2$, 1 mM $MgCl_2$, pH 7.2), then immersed 10 s in $Ca^{2+}$-free Ringer buffer containing 0.5 mg/ml poly-L-lysine. Cells were unroofed by scanning the coverslip with the tip sonicator at 10% of power under HKMgE buffer consisting of 70 mM KCl, 30 mM HEPES, 5 mM $MgCl_2$, 3 mM EGTA, pH 7.2. Unroofed cells were kept in HKMgE buffer. Fluorescent labeling of plasma membrane sheets was performed immediately after unroofing by incubating the sample with 100 nmol of TopFluor-TMR-PtdIns(4,5)$P_2$ suspended in 0.2% of absolute ethanol during 5 min, as reported in *Mueller et al., 2011*. Then, samples were extensively rinsed with HKMgE buffer and immediately imaged under the Zeiss LSM880 confocal microscope. Before addition of 1 µM of FCHo2-Alexa647, unroofed cells were rinsed with HKMgE buffer supplemented with 0.5 mg/ml of casein.

## Immunofluorescence

Supported lipid bilayers were fixed in 3.2% PFA in PBS for 2 min at room temperature, then rinsed in PBS twice. Samples were stained for the primary antibody for 45 min at room temperature in 1% BSA. Then, the secondary antibody was incubated for 45 min.

HT1080 cells were transfected with either pmCherry-C1 (empty vector), FCHo1-pmCherryC1, or FCHo2-pmCherryC1 using Lipofectamine 2000 (Thermo Fisher) according to the manufacturer's instructions. Then, plasma membrane staining of endogenous PI(4,5)$P_2$ was performed as described in *Elong Edimo et al., 2016*. Briefly, cells were fixed on ice with 3.7% formaldehyde and 0.2% glutaraldehyde for 15 min. After three washes with $NH_4Cl$, cells were incubated for 1 hr in blocking buffer (PIPES-BS, $NH_4Cl$ 50 mM, 1% lipid-free BSA, Saponin 0.05%), then incubated for 2 hr with recombinant GST-eGFP-PH-domain (PLCδ1) probe against PI(4,5)$P_2$ in PIPES-BS, 1% lipid-free BSA, Saponin 0.1% on ice. After three washes with PIPES-BS for 5 min, cells were incubated with 3.7% formaldehyde for 10 min, then 5 min at room temperature.

Finally, all samples were extensively rinsed in PBS, then in sterile water, and mounted with a Mowiol 4-88 mounting medium (Polysciences, Inc). Montage was allowed to solidify in the dark for 48 hr before microscope acquisitions.

The FCHo1-pmCherryC1 and FCHo2-pmCherryC1 vectors were a gift from Christien Merrifield (Addgene plasmid #27690 and #27686, respectively).

## Silica thin-film nanostructuration

$SiO_2$ vertical nanostructures were prepared on conventional borosilicate coverslips with precision thickness no. 1.5 (0.170 ± 0.005 mm), as previously reported (*Sansen et al., 2020*; *Zhang et al., 2020*). Briefly, Si masters were elaborated using LIL lithography as detailed in *Zhang et al., 2020* and *Zhang et al., 2019*. Polydimethylsiloxane (PDMS) reactants (90 w% RTV141A; 10 w% RTV141B from Bluesil) were transferred onto the master and dried at 70°C for 1 hr before unmolding.

Silica precursor solution was prepared by adding 4.22 g tetraethyl orthosilicate (TEOS) into 23.26 g absolute ethanol, then 1.5 g HCl (37%), and stirring the solution for 18 hr. The final molar composition was TEOS:HCl:EtOH = 1:0.7:25. All the chemicals were from Sigma. Gel films were obtained by dip-coating the coverslips with a ND-DC300 dip-coater (Nadetech Innovations) equipped with an EBC10 Miniclima device to control the surrounding temperature and relative humidity to 20°C and 45–50%, respectively. The thickness of the film was controlled by the withdrawal rate at 300 mm/min. After dip-coating, gel films were consolidated at 430°C for 5 min. Then, a new layer of the same solution was deposited under the same conditions for printing with the PDMS mold. After imprinting, the samples were transferred to a 70°C oven for 1 min and then to a 140°C oven for 2 min to consolidate the xerogel films before peeling off the PDMS mold. Finally, the sol–gel replicas were annealed at 430°C for 10 min for consolidation.

## Fluorescence microscopy

Images were acquired on a Zeiss LSM880 Airyscan confocal microscope (MRI facility, Montpellier). Excitation sources used were an argon laser for 488 nm and 514 nm, and a helium/neon laser for 633 nm. Acquisitions were performed on a ×63/1.4 objective. Multidimensional acquisitions were acquired via an Airyscan detector (32-channel GaAsP photomultiplier tube array detector).

## HS-AFM imaging

HS-AFM movies were acquired with an HS-AFM (SS-NEX, Research Institute of Biomolecule Metrology, Tsukuba, Japan) equipped with a superluminescent diode (wavelength, 750 nm; EXS 7505-B001, Exalos, Schlieren, Switzerland) and a digital high-speed lock-in Amplifier (Hinstra, Transcommers, Budapest, Hungary) detailed in *Colom et al., 2013* following the protocol detailed in *Zuttion et al., 2018*. Scanning was performed using USC-1.2 cantilevers featuring an electron beam deposition tip (NanoWorld, Neuchâtel, Switzerland) with a nominal spring constant k = 0.15 N/m, resonance frequency f(r) = 600 kHz, and quality factor Qc ≈ 2 under liquid conditions. For high-resolution imaging, the electron beam deposition tip was sharpened by helium plasma etching using a plasma cleaner (Diener Electronic, Ebhausen, Germany). Images were acquired in amplitude modulation mode at the minimal possible applied force that enables good quality of imaging under optical feedback parameters.

## Image processing and quantification

Line profiles of the fluorescence intensities were done using ImageJ (*Schneider et al., 2012*), and the kymographs were made using the Kymograph plugin (http://www.embl.de/eamnet/html/body_kymograph.html).

Protein binding was quantified by measuring the mean gray value of the protein channel that was then normalized by the mean gray value of the membrane intensity (as indicated by the TF-T-MR-PI(4,5)$P_2$ fluorescence) in the same image. Mean gray values were measured once protein binding reached the steady state, which was estimated from the binding kinetics to be <4 min. Protein binding was averaged from three experimental replicates. Mean gray values were measured using ImageJ. Concentrations and confocal parameters were kept constant between experiments and samples.

The automatic analysis of images to determine molecular clusters was performed with ImageJ (*Schneider et al., 2012*). Spots of different sizes were detected using a scale space spot detection (*Lindeberg, 1994*) and overlapping spots were merged. The LoG filter of the FeatureJ plugin (*Erik, 2020*) was used to create the scale space. Starting points were detected as local minima of the minimum projection and as minima on the smallest scale. A simplified linking scheme was applied that looks for minima along the scales for each starting point within the radius of the spot on the given scale. Two spots were merged if at least 20% of the surface of one spot is covered by the other. The noise tolerance for the spot detection was determined manually for each series of input images.

HS-AFM images were processed using Gwyddion, an open-source software for SPM data analysis, and WSxM (*Horcas et al., 2007*).

## Data representation and statistical analysis

Representation of cross-sectional analysis and recruitment curves was performed using Origin software. 3D rendering of Airyscan images was generated with the 3/4D visualization and analysis software Imaris (Oxford Instruments).

Statistical analysis was performed using the two-tailed, unpaired Welch's *t*-test, or ordinary one-way ANOVA and Dunnett's multiple comparisons test, with single pooled variance using GraphPad Prism software. In all statistics, the levels of significance were defined as *$p<0.05$, **$p<0.01$, ***$p<0.001$, and ****$p<0.0001$.

# Acknowledgements

The authors thank HT McMahon for kindly providing F-BAR domain protein constructs. C Goujon and O Moncorgé for helping in the transduction of HT1080 cells. C Holuka for assistance with plasma membrane sheets. P Maiuri for assistance in data analysis. JB Manneville for critical reading of the manuscript and discussion. S Roche, C Favard, and D Muriaux for scientific discussions. We acknowledge the imaging facility MRI, member of the national infrastructure France-BioImaging infrastructure supported by the French National Research Agency (ANR-10-INBS-04, «Investments for the future»). LP acknowledges the ATIP-Avenir program (AO-2016) and ANR-18-CE13-0015-02 for financial support. AC-G acknowledges the financial support from the European Research Council (ERC) under the European Union's Horizon 2020 research and innovation program (no. 803004). This project was supported by the LabEx NUMEV (ANR-10-LABX-0020) within the I-Site MUSE.

# Additional information

### Funding

| Funder | Grant reference number | Author |
| --- | --- | --- |
| Agence Nationale de la Recherche | ANR-10-INBS-04 | Volker Baecker |
| ATIP-Avenir | AO-2016 | Laura Picas |
| Agence Nationale de la Recherche | ANR-18-CE13-0015-02 | Laura Picas |
| European Research Council | No.803004 | Adrien Carretero-Genevrier |
| LabEx NUMEV | ANR-10-LABX-0020 | Adrian Carretero-Genevrier Laura Picas |

The funders had no role in study design, data collection and interpretation, or the decision to submit the work for publication.

### Author contributions

Fatima El Alaoui, Conceptualization, Investigation, Writing - original draft; Ignacio Casuso, Investigation; David Sanchez-Fuentes, Charlotte Arpin-Andre, Raissa Rathar, Anna Castro, Thierry Lorca, Stéphane Vassilopoulos, Methodology; Volker Baecker, Formal analysis, Software; Julien Viaud, Methodology, Writing - review and editing; Adrian Carretero-Genevrier, Methodology, Supervision; Laura Picas, Conceptualization, Formal analysis, Funding acquisition, Supervision, Writing - original draft, Writing - review and editing

### Author ORCIDs

Fatima El Alaoui ⓘ http://orcid.org/0000-0003-3298-4078
Raissa Rathar ⓘ http://orcid.org/0000-0001-8766-2186
Volker Baecker ⓘ http://orcid.org/0000-0002-9129-6403
Anna Castro ⓘ http://orcid.org/0000-0002-3655-1352

Julien Viaud http://orcid.org/0000-0003-4406-5642
Stéphane Vassilopoulos http://orcid.org/0000-0003-0172-330X
Adrian Carretero-Genevrier http://orcid.org/0000-0003-0488-9452
Laura Picas http://orcid.org/0000-0002-5619-5228

## Decision letter and Author response

Decision letter https://doi.org/10.7554/eLife.73156.sa1
Author response https://doi.org/10.7554/eLife.73156.sa2

---

# Additional files

## Supplementary files
• Transparent reporting form

## Data availability

All data generated or analyzed during this study are included in the manuscript and supporting file. Datasets are available at Dryad, https://doi.org/10.5061/dryad.n8pk0p2wp.

The following dataset was generated:

| Author(s) | Year | Dataset title | Dataset URL | Database and Identifier |
|---|---|---|---|---|
| Picas L | 2022 | Data from: Structural organization and dynamics of FCHo2 docking on membranes | http://dx.doi.org/10.5061/dryad.n8pk0p2wp | Dryad Digital Repository, 10.5061/dryad.n8pk0p2wp |

/ # References

**Abd Halim KB**, Koldsø H, Sansom MSP. 2015. Interactions of the EGFR juxtamembrane domain with PIP2-containing lipid bilayers: Insights from multiscale molecular dynamics simulations. *Biochimica et Biophysica Acta* **1850**:1017–1025. DOI: https://doi.org/10.1016/j.bbagen.2014.09.006, PMID: 25219456

**Braunger JA**, Kramer C, Morick D, Steinem C. 2013. Solid supported membranes doped with PIP2: influence of ionic strength and pH on bilayer formation and membrane organization. *Langmuir* **29**:14204–14213. DOI: https://doi.org/10.1021/la402646k, PMID: 24199623

**Chen Y**, Yong J, Martínez-Sánchez A, Yang Y, Wu Y, De Camilli P, Fernández-Busnadiego R, Wu M. 2019. Dynamic instability of clathrin assembly provides proofreading control for endocytosis. *The Journal of Cell Biology* **218**:3200–3211. DOI: https://doi.org/10.1083/jcb.201804136, PMID: 31451612

**Colom A**, Casuso I, Rico F, Scheuring S. 2013. A hybrid high-speed atomic force-optical microscope for visualizing single membrane proteins on eukaryotic cells. *Nature Communications* **4**:2155. DOI: https://doi.org/10.1038/ncomms3155, PMID: 23857417

**Daste F**, Walrant A, Holst MR, Gadsby JR, Mason J, Lee J-E, Brook D, Mettlen M, Larsson E, Lee SF, Lundmark R, Gallop JL. 2017. Control of actin polymerization via the coincidence of phosphoinositides and high membrane curvature. *The Journal of Cell Biology* **216**:3745–3765. DOI: https://doi.org/10.1083/jcb.201704061, PMID: 28923975

**Day KJ**, Kago G, Wang L, Richter JB, Hayden CC, Lafer EM, Stachowiak JC. 2021. Liquid-like protein interactions catalyse assembly of endocytic vesicles. *Nature Cell Biology* **23**:366–376. DOI: https://doi.org/10.1038/s41556-021-00646-5, PMID: 33820972

**Doherty GJ**, McMahon HT. 2009. Mechanisms of endocytosis. *Annual Review of Biochemistry* **78**:857–902. DOI: https://doi.org/10.1146/annurev.biochem.78.081307.110540, PMID: 19317650

**Elong Edimo W**, Ghosh S, Derua R, Janssens V, Waelkens E, Vanderwinden J-M, Robe P, Erneux C. 2016. SHIP2 controls plasma membrane PI(4,5)P2 thereby participating in the control of cell migration in 1321 N1 glioblastoma cells. *Journal of Cell Science* **129**:1101–1114. DOI: https://doi.org/10.1242/jcs.179663, PMID: 26826186

**Erik M**. 2020. FeatureJ: An ImageJ Plugin Suite for Image Feature Extraction. 2.0.0. Imagescience. https://imagescience.org/meijering/software/featurej/

**Frost A**, Perera R, Roux A, Spasov K, Destaing O, Egelman EH, De Camilli P, Unger VM. 2008. Structural basis of membrane invagination by F-BAR domains. *Cell* **132**:807–817. DOI: https://doi.org/10.1016/j.cell.2007.12.041, PMID: 18329367

**Godlee C**, Kaksonen M. 2013. Review series: From uncertain beginnings: initiation mechanisms of clathrin-mediated endocytosis. *The Journal of Cell Biology* **203**:717–725. DOI: https://doi.org/10.1083/jcb.201307100, PMID: 24322426

Haucke V, Kozlov MM. 2018. Membrane remodeling in clathrin-mediated endocytosis. *Journal of Cell Science* **131**:jcs216812. DOI: https://doi.org/10.1242/jcs.216812, PMID: 30177505

Henne WM, Kent HM, Ford MGJ, Hegde BG, Daumke O, Butler PJG, Mittal R, Langen R, Evans PR, McMahon HT. 2007. Structure and analysis of FCHo2 F-BAR domain: a dimerizing and membrane recruitment module that effects membrane curvature. *Structure* **15**:839–852. DOI: https://doi.org/10.1016/j.str.2007.05.002, PMID: 17540576

Henne WM, Boucrot E, Meinecke M, Evergren E, Vallis Y, Mittal R, McMahon HT. 2010. FCHo proteins are nucleators of clathrin-mediated endocytosis. *Science* **328**:1281–1284. DOI: https://doi.org/10.1126/science.1188462, PMID: 20448150

Heuser J. 2000. The production of "cell cortices" for light and electron microscopy. *Traffic* **1**:545–552. DOI: https://doi.org/10.1034/j.1600-0854.2000.010704.x, PMID: 11208142

Hollopeter G, Lange JJ, Zhang Y, Vu TN, Gu M, Ailion M, Lambie EJ, Slaughter BD, Unruh JR, Florens L, Jorgensen EM. 2014. The membrane-associated proteins FCHo and SGIP are allosteric activators of the AP2 clathrin adaptor complex. *eLife* **3**:e03648. DOI: https://doi.org/10.7554/eLife.03648, PMID: 25303366

Honigmann A, van den Bogaart G, Iraheta E, Risselada HJ, Milovanovic D, Mueller V, Müllar S, Diederichsen U, Fasshauer D, Grubmüller H, Hell SW, Eggeling C, Kühnel K, Jahn R. 2013. Phosphatidylinositol 4,5-bisphosphate clusters act as molecular beacons for vesicle recruitment. *Nature Structural & Molecular Biology* **20**:679–686. DOI: https://doi.org/10.1038/nsmb.2570, PMID: 23665582

Höning S, Ricotta D, Krauss M, Späte K, Spolaore B, Motley A, Robinson M, Robinson C, Haucke V, Owen DJ. 2005. Phosphatidylinositol-(4,5)-bisphosphate regulates sorting signal recognition by the clathrin-associated adaptor complex AP2. *Molecular Cell* **18**:519–531. DOI: https://doi.org/10.1016/j.molcel.2005.04.019, PMID: 15916959

Horcas I, Fernández R, Gómez-Rodríguez JM, Colchero J, Gómez-Herrero J, Baro AM. 2007. WSXM: a software for scanning probe microscopy and a tool for nanotechnology. *The Review of Scientific Instruments* **78**:013705. DOI: https://doi.org/10.1063/1.2432410, PMID: 17503926

Huff J, Bergter A, Birkenbeil J, Kleppe I, Engelmann R, Krzic U. 2017. The new 2D Superresolution mode for ZEISS Airyscan. *Nature Methods* **14**:1223–1229. DOI: https://doi.org/10.1038/nmeth.f.404

Itoh T, Erdmann KS, Roux A, Habermann B, Werner H, De Camilli P. 2005. Dynamin and the actin cytoskeleton cooperatively regulate plasma membrane invagination by BAR and F-BAR proteins. *Developmental Cell* **9**:791–804. DOI: https://doi.org/10.1016/j.devcel.2005.11.005, PMID: 16326391

Kadlecova Z, Spielman SJ, Loerke D, Mohanakrishnan A, Reed DK, Schmid SL. 2017. Regulation of clathrin-mediated endocytosis by hierarchical allosteric activation of AP2. *The Journal of Cell Biology* **216**:167–179. DOI: https://doi.org/10.1083/jcb.201608071, PMID: 28003333

Koldsø H, Shorthouse D, Hélie J, Sansom MSP. 2014. Lipid clustering correlates with membrane curvature as revealed by molecular simulations of complex lipid bilayers. *PLOS Computational Biology* **10**:e1003911. DOI: https://doi.org/10.1371/journal.pcbi.1003911, PMID: 25340788

Krauss M, Kukhtina V, Pechstein A, Haucke V. 2006. Stimulation of phosphatidylinositol kinase type I-mediated phosphatidylinositol (4,5)-bisphosphate synthesis by AP-2mu-cargo complexes. *PNAS* **103**:11934–11939. DOI: https://doi.org/10.1073/pnas.0510306103, PMID: 16880396

Lehmann M, Lukonin I, Noé F, Schmoranzer J, Clementi C, Loerke D, Haucke V. 2019. Nanoscale coupling of endocytic pit growth and stability. *Science Advances* **5**:eaax5775. DOI: https://doi.org/10.1126/sciadv.aax5775, PMID: 31807703

Lemmon MA, Ferguson KM, O'Brien R, Sigler PB, Schlessinger J. 1995. Specific and high-affinity binding of inositol phosphates to an isolated pleckstrin homology domain. *PNAS* **92**:10472–10476. DOI: https://doi.org/10.1073/pnas.92.23.10472, PMID: 7479822

Lindeberg T. 1994. Scale-Space Theory in Computer Vision. Boston, MA: Springer US. DOI: https://doi.org/10.1007/978-1-4757-6465-9

Ma L, Umasankar PK, Wrobel AG, Lymar A, McCoy AJ, Holkar SS, Jha A, Pradhan-Sundd T, Watkins SC, Owen DJ, Traub LM. 2016. Transient Fcho1/2·Eps15/R·AP-2 Nanoclusters Prime the AP-2 Clathrin Adaptor for Cargo Binding. *Developmental Cell* **37**:428–443. DOI: https://doi.org/10.1016/j.devcel.2016.05.003, PMID: 27237791

Mim C, Cui H, Gawronski-Salerno JA, Frost A, Lyman E, Voth GA, Unger VM. 2012. Structural basis of membrane bending by the N-BAR protein endophilin. *Cell* **149**:137–145. DOI: https://doi.org/10.1016/j.cell.2012.01.048, PMID: 22464326

Mueller V, Ringemann C, Honigmann A, Schwarzmann G, Medda R, Leutenegger M, Polyakova S, Belov VN, Hell SW, Eggeling C. 2011. STED nanoscopy reveals molecular details of cholesterol- and cytoskeleton-modulated lipid interactions in living cells. *Biophysical Journal* **101**:1651–1660. DOI: https://doi.org/10.1016/j.bpj.2011.09.006, PMID: 21961591

Mulkearns EE, Cooper JA. 2012. FCH domain only-2 organizes clathrin-coated structures and interacts with Disabled-2 for low-density lipoprotein receptor endocytosis. *Molecular Biology of the Cell* **23**:1330–1342. DOI: https://doi.org/10.1091/mbc.E11-09-0812, PMID: 22323290

Picas L, Carretero-Genevrier A, Montero MT, Vázquez-Ibar JL, Seantier B, Milhiet P-E, Hernández-Borrell J. 2010. Preferential insertion of lactose permease in phospholipid domains: AFM observations. *Biochimica et Biophysica Acta* **1798**:1014–1019. DOI: https://doi.org/10.1016/j.bbamem.2010.01.008, PMID: 20096263

Picas L, Viaud J, Schauer K, Vanni S, Hnia K, Fraisier V, Roux A, Bassereau P, Gaits-Iacovoni F, Payrastre B, Laporte J, Manneville J-B, Goud B. 2014. BIN1/M-Amphiphysin2 induces clustering of phosphoinositides to

recruit its downstream partner dynamin. *Nature Communications* **5**:5647. DOI: https://doi.org/10.1038/ncomms6647, PMID: 25487648

**Posor Y**, Eichhorn-Grünig M, Haucke V. 2015. Phosphoinositides in endocytosis. *Biochimica et Biophysica Acta* **1851**:794–804. DOI: https://doi.org/10.1016/j.bbalip.2014.09.014, PMID: 25264171

**Ramesh P**, Baroji YF, Reihani SNS, Stamou D, Oddershede LB, Bendix PM. 2013. FBAR syndapin 1 recognizes and stabilizes highly curved tubular membranes in a concentration dependent manner. *Scientific Reports* **3**:1565. DOI: https://doi.org/10.1038/srep01565, PMID: 23535634

**Sansen T**, Sanchez-Fuentes D, Rathar R, Colom-Diego A, El Alaoui F, Viaud J, Macchione M, de Rossi S, Matile S, Gaudin R, Bäcker V, Carretero-Genevrier A, Picas L. 2020. Mapping Cell Membrane Organization and Dynamics Using Soft Nanoimprint Lithography. *ACS Applied Materials & Interfaces* **12**:29000–29012. DOI: https://doi.org/10.1021/acsami.0c05432, PMID: 32464046

**Schneider CA**, Rasband WS, Eliceiri KW. 2012. NIH Image to ImageJ: 25 years of image analysis. *Nature Methods* **9**:671–675. DOI: https://doi.org/10.1038/nmeth.2089, PMID: 22930834

**Sheetz MP**. 2001. Cell control by membrane-cytoskeleton adhesion. *Nature Reviews. Molecular Cell Biology* **2**:392–396. DOI: https://doi.org/10.1038/35073095, PMID: 11331914

**Sochacki KA**, Dickey AM, Strub MP, Taraska JW. 2017. Endocytic proteins are partitioned at the edge of the clathrin lattice in mammalian cells. *Nature Cell Biology* **19**:352–361. DOI: https://doi.org/10.1038/ncb3498, PMID: 28346440

**Su M**, Zhuang Y, Miao X, Zeng Y, Gao W, Zhao W, Wu M. 2020. Comparative Study of Curvature Sensing Mediated by F-BAR and an Intrinsically Disordered Region of FBP17. *IScience* **23**:101712. DOI: https://doi.org/10.1016/j.isci.2020.101712, PMID: 33205024

**Taylor MJ**, Perrais D, Merrifield CJ. 2011. A high precision survey of the molecular dynamics of mammalian clathrin-mediated endocytosis. *PLOS Biology* **9**:e1000604. DOI: https://doi.org/10.1371/journal.pbio.1000604, PMID: 21445324

**Umasankar PK**, Sanker S, Thieman JR, Chakraborty S, Wendland B, Tsang M, Traub LM. 2012. Distinct and separable activities of the endocytic clathrin-coat components Fcho1/2 and AP-2 in developmental patterning. *Nature Cell Biology* **14**:488–501. DOI: https://doi.org/10.1038/ncb2473, PMID: 22484487

**van den Bogaart G**, Meyenberg K, Risselada HJ, Amin H, Willig KI, Hubrich BE, Dier M, Hell SW, Grubmüller H, Diederichsen U, Jahn R. 2011. Membrane protein sequestering by ionic protein-lipid interactions. *Nature* **479**:552–555. DOI: https://doi.org/10.1038/nature10545, PMID: 22020284

**Walrant A**, Saxton DS, Correia GP, Gallop JL. 2015. Triggering actin polymerization in *Xenopus* egg extracts from phosphoinositide-containing lipid bilayers. *Methods in Cell Biology* **128**:125–147. DOI: https://doi.org/10.1016/bs.mcb.2015.01.020, PMID: 25997346

**Wang Y**, Gao J, Guo X, Tong T, Shi X, Li L, Qi M, Wang Y, Cai M, Jiang J, Xu C, Ji H, Wang H. 2014. Regulation of EGFR nanocluster formation by ionic protein-lipid interaction. *Cell Research* **24**:959–976. DOI: https://doi.org/10.1038/cr.2014.89, PMID: 25001389

**Wu M**, Huang B, Graham M, Raimondi A, Heuser JE, Zhuang X, De Camilli P. 2010. Coupling between clathrin-dependent endocytic budding and F-BAR-dependent tubulation in a cell-free system. *Nature Cell Biology* **12**:902–908. DOI: https://doi.org/10.1038/ncb2094, PMID: 20729836

**Zewe JP**, Miller AM, Sangappa S, Wills RC, Goulden BD, Hammond GRV. 2020. Probing the subcellular distribution of phosphatidylinositol reveals a surprising lack at the plasma membrane. *The Journal of Cell Biology* **219**:6127. DOI: https://doi.org/10.1083/jcb.201906127, PMID: 32211893

**Zhang Q**, Sánchez-Fuentes D, Gómez A, Desgarceaux R, Charlot B, Gàzquez J, Carretero-Genevrier A, Gich M. 2019. Tailoring the crystal growth of quartz on silicon for patterning epitaxial piezoelectric films. *Nanoscale Advances* **1**:3741–3752. DOI: https://doi.org/10.1039/C9NA00388F

**Zhang Q**, Sánchez-Fuentes D, Desgarceaux R, Escofet-Majoral P, Oró-Soler J, Gázquez J, Larrieu G, Charlot B, Gómez A, Gich M, Carretero-Genevrier A. 2020. Micro/Nanostructure Engineering of Epitaxial Piezoelectric α-Quartz Thin Films on Silicon. *ACS Applied Materials & Interfaces* **12**:4732–4740. DOI: https://doi.org/10.1021/acsami.9b18555, PMID: 31880913

**Zhao H**, Michelot A, Koskela EV, Tkach V, Stamou D, Drubin DG, Lappalainen P. 2013. Membrane-sculpting BAR domains generate stable lipid microdomains. *Cell Reports* **4**:1213–1223. DOI: https://doi.org/10.1016/j.celrep.2013.08.024, PMID: 24055060

**Zuttion F**, Redondo-Morata L, Marchesi A, Casuso I. 2018. High-Resolution and High-Speed Atomic Force Microscope Imaging. *Methods in Molecular Biology* **1814**:181–200. DOI: https://doi.org/10.1007/978-1-4939-8591-3_11, PMID: 29956233

