## [Decision Letter]

[Editors' note: this paper was reviewed by Review Commons.]

---

## [Author Response]

We thank the reviewers for their encouraging feedback and constructive reports on our manuscript. We are also pleased the read that reviewers #1 and #3 found this work of interest and provide novel insights in the endocytosis field.

We have carefully considered and addressed the questions and comments raised by the three reviewers. Accordingly, we have updated the text and figures and included new analyses in the revised version of the manuscript. Finally, we have performed additional experiments to address the comments raised by the referees.

Reviewer #1:Major points:1. The paper nicely confirms that PI4,5P2 is an important element that directs the recruitment of FCHo2 protein to membranes, consistent with earlier experiments in vitro and in living cells. The most novel aspect of this study pertains to the observations in Figure 2 showing that FCHo2 application directly or indirectly induces PI4,5P2 clustering. Hence, FCHo2 and PI4,5P2 may be part of a positive feedback loop. When conducted in the presence of cytoplasm co-clustering of PI4,5P2 with FCHo2 is seen to facilitate clathrin recruitment. The grand problem with these data is that it remains largely unclear which of these effects are a direct consequence of FCHo2 binding to PI4,5P2, its ability to self-assemble, or association with other endocytic proteins present in cytoplasmic extracts as well as with receptors contained on plasma membrane sheets such as the EGFR. The latter scenario is supported by the observation that while association of FCHo2 with TfRs causes PI4,5P2 redistribution binding of FCHo2 to EGFR spots appears to result in a rise in PI4,5P2 intensity, in addition to the formation of clusters. Hence, this may reflect activation of PIPK typeI by FCHo2 (for example by binding AP2 which in turn can recruit or activate PIPK typeI present on plasma membrane sheets) or other mechanisms such as EGFR activation induced phosphoinositide signaling. This should be addressed experimemtally.

The reviewer brings in an excellent remark. In our experiments, we started analyzing the effect of FCHo2 in PI(4,5)P2 in plasma membrane sheets. In this case, we performed experiments without cytoplasmic extracts (i.e., absence of other endocytic proteins except for recombinant FCHo2). Although we expect to remove any cytosolic protein during the unroofing process and wash-outs of cells, we could not exclude the presence of resident type I PIPK at the plasma membrane. For this reason, we conducted experiments on plasma membrane sheets in the absence of ATP. Thus, the PI4,5P2 redistribution that we observe on plasma membrane sheets appears a direct effect of the FCHo2 binding and is not due to the transformation of PI(4)P to PI(4,5)P2.

To evaluate the contribution of other endocytic proteins, we decided to simplify the system with lipid bilayers to prevent, for instance, EGFR activation-induced phosphoinositide signaling, as pointed out by the reviewer. We used cytosolic extracts since producing recombinant FCHo2 partners, specially AP2 is technically challenging. We also considered depleting AP2 or PIPK Iγ from X. egg extracts, but this strategy was out of our competencies. A feasible strategy would be to inhibit PIPK Iγ from extracts, but we failed to find a commercial inhibitor. To tackle this technical issue, we decided to use PS-containing membranes as a control in which full-length FCHo2 still binds to the membrane (i.e., its functional domains are accessible to interact with partners). Also, we used the F-BAR domain alone in PI(4,5)P2-containing membranes as a strategy to preserve the formation of PI(4,5)P2 clustering (Zhao et al. Cell Reports, 2013) but, in principle, prevent domain-interactions with FCHo2 partners (e.g., mediated by the C-terminal by a u-homology domain or AP2 Activator (APA) domain).

Our results showed that PI(4,5)P2 clustering is present in the absence of cytosolic extracts and ATP and that this effect is enhanced in cytoplasmic extracts even in the case of the F-BAR domain alone, which point out that FCHo2-mediated PI4,5P2 clustering is well a direct effect, but it is indeed potentiated by partners, as showed when using the F-BAR-x construct from Henne et al., in agreement with the role of AP2 (Krauss et al. 2006) and the stabilization of an endocytic network in the formation of clathrin-coated structures reported by Ma et al. Dev. Cell, 2016 and Lehmann et al. Science Adv., 2019.

Accordingly, we have better explained these points in the revised version of the manuscript.

2. In my view the present set of experiments would need to be complemented by careful structure-function analysis using available and well-characterized FCHo2 mutants (e.g. mutation of basic residues shown to be required for PI4,5P2 binding, self-assembly mutants) or chimeric proteins to distinguish direct effects of FCHo2 binding to PI4,5P2 from indirect ones. Moreover, only part of the data presented are compelling in my view: For example, I see no evidence for PI4,5P2 clustering upon addition of FCHo2 in the images shown in Figure 1B or G, in contrast to the results shown in Figure 2 and the central claims of the manuscript. Likewise, the claimed exclusion of the FBAR domain from lipid nanodomain pillars in Figure 5B is not overt to me, nor is the reason for this behavior.

The suggestion of the reviewer to complement experiments with structure-function analysis using FCHo2 mutants is a good one, and we already thought about this. The reviewer is correct that mutants have been characterized, especially in living cells. Mutants showing a dimerization (F38+W73E and F38E+W73E) and membrane binding defect (K146E+K163 and K146E and K165E) do not associate with membranes (Henne et al. Science, 2010 and Lehmann et al. Science Adv., 2019). Because these mutants are predominantly cytosolic, it is unclear how they will decouple the different functions of FCHo2 on our lipid bilayers or PM sheets assay. To tackle this limitation, and following up with point 1 above, our strategy was to use the F-BAR domain alone or to tune the lipid composition of supported bilayers, which appeared as a feasible strategy to address the direct effect of FCHo2 in PI(4,5)P2 reorganization. Other interesting FCHo2 mutants and that, in this case, associate with membranes are the I268N and L136E, which are reported in autoimmune diseases (Henne et al. Science, 2010). However, disease-like mutants are out of the scope of this work, and generating their recombinant version would require careful validation of protein folding, labeling, and membrane interaction in vitro that we are afraid will not be feasible due to time and resource limitations. However, the F-BAR-x construct (F-BAR domain + polybasic motif, from Henne et al. Science, 2010) has been validated in vitro. The F-BAR-x construct (aa 1-430) is similar to the F-BAR-APA ( F-BAR + AP2 Activator domain, aa 1-394) from Lehmann et al. 2019 and should indeed address the points raised by the reviewer regarding the cytosolic extracts (point 1, already addressed above) and the exclusion of the F-BAR domain from nano-domes (point 2, see below).

As pointed by the reviewer, the fact that the F-BAR domain is not accumulated on nano-domes, as compared to the full-length protein, might reflect a similar behavior to that reported in the case of the F-BAR domain of FBP17 (Su et al. iScience, 2020). In this work, they found that the curvature sensing abilities of the protein do not exclusively rely on the F-BAR domain but the presence of intrinsically disordered regions. We believe that this is an interesting and important point to address, and thus, we have perform additional experiments on nano-domes using the F-BAR-x construct reported in Henne et al. 2010. (which contains polybasic region interacting with PI(4,5)P2 following the FBAR domain but lacks part of the central disordered region and the c-terminal uHD). Our results show that neither the F-BAR domain nor the F-BAR-x accumulate at the base of nano-domes as compared to the full-length protein. This observation suggests that, in addition to PI(4,5)P2 binding, the ability of FCHo2 to accumulate on curvatures of radius, R ~ 150 nm, might be encoded by the protein regions downstream residue 430. Thanks to the comment raised by the reviewer, we believe that elucidating the role of these regions is an interesting point that we would like to perform in the future, as it will require generating new mutants and validate their functionality as recombinant proteins in vitro (which, we believe is a complete project/work in itself).

Finally, we thank the reviewer for bringing up that Figure 1B or G (Figure S3 and S5 in the revised version of the manuscript) might be confusing. The main objective of Figure 1 was to validate that FCHo2 is functional and can spatially recognize PI(4,5)P2 on flat membranes. To this aim, in Figure 1 we chose images that highlight the interaction of FCHo2 with pre-existing PI(4,5)P2 domains along with the quantification of this type of event. The quantification of the contribution of FCHo2 on PI(4,5)P2 reorganization is presented later on in Figure 2. Following up with the point raised by this reviewer and also, as highlighted by reviewer #2 (point 2), we realized that presenting two different figures 1 and 2 in the main manuscript is redundant and misleading in what concerns one of the main novelties/messages of the present work, which is that FCHo2 is an actuator of PI(4,5)P2 clustering formation and that this effect is enhanced in the presence of partners. Accordingly, we have amended the order of figures to clarify the main findings of the present work.

3. This brings me to a third major point: Both PI4,5P2 binding as well as self-assembly of FCHo2 are largely encoded within the FBAR domain. In spite of this it appears that the FBAR domain alone in spite of its ability to cluster PI4,5P2 (see Figure 2F) and to form molecular clusters as assessed by AFM it fails to polymerize onto curved membrane-coated nanopillars unlike the full-length protein. This conundrum remains unexplained and may relate to the recently claimed ability of FCHo2 to phase separate via its intrinsically disordered region. Hence, experiments addressing this important issue would significantly increase the impact of the present dataset.

The reviewer brings up an essential point that we agree needs to be addressed, in line with point 2 (detailed above). Accordingly, we have performed new experiments on nano-domes by producing a recombinant version of the F-BAR-x construct reported in Henne et al. Science, 2010. (which contains polybasic region interacting with PI(4,5)P2 following the F-BAR domain but lacks part of the central disordered region and the c-terminal uHD).

4. Although not necessary for publication in a scientific journal in general, evidence that FCHo2 can induce PI4,5P2 clustering in living cells would raise the impact of the study considerably.

We agree with the reviewer that confirmation of FCHo2-mediated PI(4,5)P2 clustering formation in living cells would be a relevant point to show for the present manuscript. However, we want to highlight that following the dynamics of phosphoinositides in living cells is not a trivial issue. Thus, phosphoinositide detection is often limited to fixed samples (Idevall-Hagren and De Camilli. BBA, 2015). We are aware of a recent strategy that would prevent the competing effects of phosphoinositide-binding domains (over)expression, which consists in combining phosphoinositide lipid dyes (as we used in our in vitro systems) with orthogonal approaches in mammalian cells (Zewe et al. JCB, 2020). However, setting up this type of assay in the framework of the present work will not be feasible due to resource and time limitations. Thus, to address the point raised by the reviewer, we have performed new experiments to correlate the expression of FCHo 1 and 2 with PI(4,5)P2 intensity using a recombinant version of a GFP-PH(PLCd1) to detect the endogenous localization of PI(4,5)P2 at the plasma membrane of fixed cells.

Minor comments:1. The statistical basis for the various experimental datasets, in particular the definition of n and the exact p values should be spelled out in the figure legends.

We thank the reviewer for this remark. We have included the definition of n and a detailed explanation of the statistical tests in the methods section.

However, while we can get the exact p-value for tests reporting a P > 0.0001 with the software GraphPad Prism, we failed to obtain the exact value on tests with P < 0.0001, which is the case in Figure 1G-F.

2. A recent study by Lehmann et al. in Science Advances has provided important molecular leads regarding the role of FCHo2 as an important regulator of clathrin-coated pit size and stability via the formation of ring-like molecular assemblies. This paper should be quoted and discussed.

We thank the reviewer for bringing up the study by Lehmann et al. Science Adv. 2019, which is highly pertinent to support that FCHo2 ring-like organizations exist in cells. Also, this study reports that both the F-BAR and interacting partners are required for CCP size and stability, which is also relevant for interpreting our results (e.g., experiments with cytosolic extracts). Accordingly, the paper is now quoted and discussed in the revised version of the manuscript.

Reviewer #2:1) Abstract. "bottom-up synthetic approaches". Introduction, "sets the load of…", results, "freshly prepared energy mix", please use plain language.

This remark is now amended in the revised version of the manuscript.

2) Figure 1A-E, membrane fragments (sonicated unroofed cells) with PI(4,5)P2 recruit FCHo2. This is well understood. As is the observation (Figure 1F), that in supported bilayers, PIP4,5P2 recruits FCHo. It is also expected that FCHo punctae would have a longer, more stable association with clustered PI4,5P2.Might be redundant and that indeed, Figure 1 rather reports a validation of the functionality of fcho and its abiity to interact with pip2.

The main objective of figure 1 was to show that recombinant FCHo2 is functional and can spatially recognize PI(4,5)P2 on flat membranes in vitro. Therefore, we considered that providing this evidence was necessary before addressing its role in PI(4,5)P2 clustering formation. However, we agree that this might lead to redundancy between figures 1 and 2 and a less clear appreciation of the main findings of the manuscript, as also pointed out by reviewer #1 (point 2). Accordingly, figure 1 is now part of the supplementary data, and we have amended the text and figures in the revised version of the manuscript.

3) Figure 2A, "Injection of FCHo2-Alexa647 on TfR-GFP plasma membrane sheets led to the binding of the protein and the formation of sub-micrometric FCHo2-positive puncta that co-localized with PI(4,5)P2 and the cargo receptor (Figure 2A"). I assume by the cargo receptor, the authors are referring to the transferrin receptor. There does not appear to be TfR enrichment if the FCHo2 punctae. Yet there does appear to be a relatively broad change in TfR distribution.

The reviewer is correct that the description of Figure 2A (Figure 1B in the new version) needs clarification. Accordingly, we have amended this part in the revised version of the manuscript. Indeed, “cargo receptor” refers to the TfR as our initial studies to investigate the role of FCHo2 on PI(4,5)P2 clustering formation were focused on this receptor as a hallmark of the clathrin pathway (e.g., Taylor et al. Plos Biology 2012) and based on the work of Henne et al. Science 2010. However, in the absence of ATP and other endocytic proteins, we found that the effect of FCHo2 was more pronounced on EGFR-positive events than the TfR. This difference might be explained because both FCHo2 and EGFR share a preferential interaction for PI(4,5)P2. On supported lipid bilayers, we also observed this synergistic effect in the presence of cytosolic extracts and ATP. Thus, supporting that, although FCHo2-mediated PI(4,5)P2 clustering is independent of PI(4,5)P2 production, it appears enhanced in the presence of partners.

4) For the experiments in figures 3A/B, while airyscan does give enhanced resolution, these experiments would be better served by the use of super resolution microscopy. In fact the experiments in figures 3C-E indicate that clusters are sub 50 nm resolution.

STED not feasible

The objective of Figure 3A-B was to compare the size distribution of FCHo2 spots on cellular with in vitro membranes, in which the main advantage is that we can tune the lipid composition. These experiments were performed in unfixed samples to avoid potential differences due to protein cross-linking, especially on supported lipid bilayers. We agree with the reviewer that we could improve the spatial resolution of experiments with super-resolution microscopy*.* In the case of unfixed samples, we might envision performing STED or SIM. However, at present, this type of experiment is not technically possible since STED and SIM setups are unfortunately not available on site. This issue is particularly limiting in unfixed unroofed cells as they are prepared in the lab and imaged within 5 minutes after preparation to preserve the plasma membrane integrity. For this reason, we considered a sub-diffraction microscopy approach such as Airyscan, which is available on site, as a good compromise between lateral resolution and the ability to image “live” samples within a fast time window after preparation.

Reviewer #3:

Major comments:1) The use of fluorescence correlation in figures 1E and 1I is not appropriate and the graphs are not very convincing. A measurement of co-localisation between bright lipid and protein spots, or a quantification akin to what is presented in figure 2G would possibly be more meaningful. If matlab is available to the authors, using the CMEanalysis script from the Danuser lab (Aguet, Dev cell 2013) could potentially provide an elegant visualisation of the results.

Although figures 1E and 1I have been moved to the supplementary section (Figure S4), we appreciate the reviewer for bringing up this remark. Indeed, it is a good point since the CMEanalysis script from the Danuser lab is freely available and would certainly improve data representation. Unfortunately, for the time being, we are not familiar with matlab code. Nevertheless, following the work of Aguet, Dev cell 2013, we have represented the frequency distribution of the intensity of FCHo2 and F-BAR puncta relative to the intensity of different PI(4,5)P2 domain populations (Figure S4). We hope that the reviewer will find the new representation more appropriate to discarnate that increased intensity of PI(4,5)P2 spots is associated with an increased intensity of FCHo2 puncta.

2) The HS-AFM experiments shown in figure 4 require extra quantification (and/or examples) to substantiate the claims of FCHo2 ring formation. This is especially important considering that the centre of FCHo2 rings seems to have negative curvature (darker AFM signal) rather than the expected positive curvature found in CME pits (figure 4C).

Following the reviewer's recommendation, we have now included in Figure S8 a series of examples to support FCHo2 ring formation.

The point raised concerning the darker AFM signal at the center of the ring in specific HS-AFM frames is interesting. However, to determine whether FCHo2 might induce negative curvature at the center of the ring would require further experiments out of the scope of the present manuscript. But, we might undoubtedly envision to perform in a further work. For instance, we could not exclude that this effect might reflect transient lipid reorganizations due to a preferential interaction of FCHo2 with negatively charged lipids or, eventually, interactions of hydrophobic motifs.

3) It is not clear for me how the authors could make the height calculations shown in figure 5. The Zeiss Airyscan microscope has an axial resolution of 350 nm (Huff, Nat Meth 2017) and their dome has a radius of 150 nm. The methods section on these experiments cites a previous paper from the same group where they measure larger nanopillars (>400 nm), where these measurements are indeed perfectly feasible. If FCHo2 is indeed partitioning to flat membrane regions, it should be possible to detect its enrichment on the edge of domes using xy data alone, as the Airyscan can achieve a 120 nm lateral resolution.

The recommended z-step resolution of Airyscan LSM880 using a 63x/1.4 objective is z = 0.18 mm. The reviewer is right in his/her appreciation that on nano-domes of radius 0.15 mm and aspect ratio 1:1, the calculations that we made to estimate the maximal average height, as we reported in Sansen et al. 2020, might not be appropriated. Indeed, it is an excellent remark, and we thank the reviewer for it and the recommendation to use the xy resolution to report the enrichment of proteins on nano-domes. Accordingly, we have updated the data representation in Figure 5.

Minor comments:In my opinion, the use of the term "structure" in the title is not appropriate. This term suggests the use of structural biology techniques to define atomic coordinates or the overall shape of the molecule.

The reviewer is right and we thank him/her for the suggestion. Accordingly, we have changed the title of the manuscript to “Structural organization and dynamics of FCHo2 docking on membranes”.

1) The authors have performed various experiments comparing F-BAR alone with full length FCHo2. Explanations/hypotheses for the differences found should be discussed.

An explanation of this has now been included in the revised version of the manuscript.

2) As the FCHo2 F-BAR is not described to bind any endocytic protein, how does the authors explain the increased clathrin recruitment shown in figures 2E,F and G by the F-BAR alone.

The reviewer is right that this point needs clarification, and it is somehow related to the precedent point raised by the reviewer. Furthermore, this issue has also been brought up by reviewer #1.

Both the membrane activity (PI(4,5)P2 binding and clustering formation) and FCHo2 network interaction with binding partners are required to direct clathrin-coated structures' growth (as reported by Lehmann et al. 2019). By comparing the F-BAR and full-length protein, we aimed to uncouple the contribution of PI(4,5)P2 clustering from the requirement of FCHo2 to interact with binding partners such as AP2, which might activate type I PIPK (i.e., de novo production), or other endocytic proteins such as Epsin2 (for instance, via Eps15), which is also known to induce PI(4,5)P2 clustering (Picas et al. 2014). The fact that the F-BAR domain alone can increase clathrin recruitment suggests that FCHo2 directly promotes pre-endocytic events via PI(4,5)P2-rich interfaces, for instance, by increasing the stability of the endocytic network through avidity. Furthermore, the fact that FCHo2 effect is enhanced with partners is likely to indicate that PI(4,5)P2 clustering might act as a complementary or synergistic mechanism to PI(4,5)P2 synthesis.

We have included an explanation in the manuscript on the differences between the F-BAR domain and full-length protein concerning the increased effect on the appearance of clathrin structures.

3) The authors should clearly state that the F-BAR domain used does not contain the polybasic motif found in F-BAR-x (Henne, Science 2010).

Although we have now included additional experiments using the F-BAR-x construct found in Henne et al. following the recommendations of reviewer #1, we have also included a statement on the differences of the domains that were used in the revised version of the manuscript.

4) The authors cite FPB17 F-BAR constructs in the methods section. They are not used in the manuscript.

We thank the reviewer for bringing up this mistake. Accordingly, the construct has been removed from the methods section.